# Three rate-determining protein roles in photosynthetic $O_2$-evolution addressed by time-resolved experiments on genetically modified photosystems

Sarah M. Mäusle [1,5], Gianluca Parisse [2,5], Ricardo Assunção[1], Cristina De Santis [2], Philipp S. Simon [1,3], Daniele Narzi [2], Leonardo Guidoni[2] ✉, Richard J. Debus [4] ✉ & Holger Dau [1] ✉

Light-driven water splitting by plants, algae and cyanobacteria is pivotal for global bioenergetics and biomass formation. A manganese cluster bound to the photosystem II proteins catalyzes the complex reaction at high rate, but the rate-determining factors are insufficiently understood. Here we trace the oxygen-evolution transition by time-resolved polarography and infrared spectroscopy for cyanobacterial photosystems genetically modified at two strategic sites, complemented by computational chemistry. Our results highlight three rate-determining roles of the protein environment of the metal cluster: acceleration of proton-coupled electron transfer, acceleration of substrate-water insertion after $O_2$-formation, and balancing of rate-determining enthalpic and entropic contributions. Whereas in general the substrate-water insertion step may be unresolvable in time-resolved experiments, here it likely becomes traceable because of deceleration by genetic modification. Our results may stimulate new time-resolved experiments on substrate-water insertion in photosynthesis, clarification of enthalpy-entropy compensation in enzyme catalysis, and knowledge-guided development of inorganic catalyst materials.

Plants, algae, and cyanobacteria harvest solar energy and convert it into chemical energy by producing carbohydrates from water and carbon dioxide, coupled with the release of molecular oxygen ($O_2$) into the atmosphere. This process of oxygenic photosynthesis facilitates primary biomass formation and provides the chemically stored energy that "fuels" life on Earth. A protein-cofactor complex called photosystem II (PSII) plays a central role by facilitating the light-driven oxidation of two "substrate water" molecules, thereby generating $O_2$ as well as four energized electrons (reducing equivalents) and four protons[1–5]. Here, we address the clearly crucial but still insufficiently understood roles of H-bonded protein-bound water clusters in photosynthetic oxygen evolution. Progress comes from the informative combination of problem-adapted methodology: site-directed mutagenesis verified by protein mass spectroscopy, infrared (IR) difference spectroscopy in various frequency ranges, activation energy analyses based on time-resolved $O_2$-polarography, tracking of reaction sequences by time-resolved IR spectroscopy, and large-scale molecular dynamics (MD) simulation

[1]Department of Physics, Freie Universität Berlin, Berlin, Germany. [2]Department of Physical and Chemical Sciences, University of L'Aquila, L'Aquila, Italy. [3]Molecular Biophysics and Integrated Bioimaging Division, Lawrence Berkeley National Laboratory, Berkeley, USA. [4]Department of Biochemistry, University of California, Riverside, USA. [5]These authors contributed equally: Sarah M. Mäusle, Gianluca Parisse. ✉e-mail: leonardo.guidoni@univaq.it; richard.debus@ucr.edu; holger.dau@fu-berlin.de

with a force-field specifically parameterized for the PSII state prior to onset of $O_2$-formation.

The absorption of four light quanta by the PSII pigments is required to complete one turnover of the water-oxidation cycle, each initiating the primary charge separation and resulting in the formation of a chlorophyll cation radical ($P680^+$) at the electron donor side of PSII. The chlorophyll radical oxidizes a redox-active tyrosine residue ($Y_Z$), followed by oxidation of the protein-bound $Mn_4CaO_{5-6}$ cluster that facilitates the water oxidation chemistry in conjunction with its protein-water environment[1–5]. Figure 1 shows the sequence of events in water oxidation and its atomistic key players. The basic model proposed by Kok already in 1970 comprises four semi-stable $S_i$-states ($S_0–S_3$, with $S_1$ being the dark-stable S-state) and the transiently formed $S_4$-state, where the respective subscript indicates the number of accumulated oxidizing equivalents (Fig. 1a, inner circle)[6]. Whereas in the three S-state transition leading from the $S_0$ to $S_3$, oxidizing equivalents are accumulated by oxidation of Mn ions[1,7–9], the $S_4$-state is likely rather reached by oxidation of an oxygen ($O_6$), resulting in a Mn-bound oxyl radical[10].

Water oxidation in PSII cannot be understood without considering the four protons that need to be removed from two water molecules prior to $O_2$ formation. Therefore, Kok's reaction cycle model has been extended to include four proton-removal steps (Fig. 1a, outer circle)[11], which mostly can be followed in time-resolved experiments (see time constants indicated in Fig. 1a)[12,13]. The binding of the two water molecules that represent the substrates of the water oxidation chemistry is also indicated in the extended S-state cycle of Fig. 1a, as established primarily by advanced $H_2O^{16}/H_2O^{18}$ water exchange experiments[3,14]. Here we focus on the oxygen-evolving $S_3 \rightarrow S_4 \rightarrow S_0$ transition, for which we have previously inferred the following sequence of events from experimental and computational results[10]:

(1) The light-induced $Y_Z^{ox}$ oxidation promotes proton release from the proton-sharing D2-E312–D1-E65 dyad within about 300 μs, $Y_Z^{ox} S_3^+$  $Y_Z^{ox} S_3^n + H^+$.

(2) Electron transfer from the $Mn_4CaO_6$-cluster to $Y_Z^{ox}$ is coupled to proton transfer (PT) to D1-D61 and results in oxyl-radical formation within about 3 ms, $Y_Z^{ox} S_3^n \rightarrow Y_Z S_4^+$.

(3) Not rate-limiting and thus kinetically invisible (i) peroxide formation, (ii) $O_2$-formation, and (iii) $O_2$-release result in a $Mn_4CaO_4$-cluster with vacant water binding sites, $S_4^+ \rightarrow S_0^+ + O_2$.

(4) Deprotonation of a water molecule and hydroxide insertion in the $Mn_4CaO_4^-$ cluster results in the stable $S_0$-state of the $Mn_4CaO_5$-cluster, $S_0^+ + H_2O \rightarrow S_0^n + H^+$, which seems to be a kinetically invisible step in the wild-type PSII[15] but possibly resolvable in the here investigated D1-N298A variant.

The $Mn_4CaO_{4-6}$ cluster is connected to the thylakoid lumen by extensive networks of H-bonded water molecules and amino-acid sidechains (Fig. 1b), which have been termed "channels". The three major channels ("Cl1", "O4", and "O1") likely facilitate proton egress and/or substrate water access[16–24]. The "Cl1" (or "broad") channel leads from $Y_Z$ and the $Mn_4CaO_{5-6}$ cluster's Ca ion past aspartate-61 of the D1 subunit (D1-D61) and the $Cl^-$ ion that is coordinated by lysine-317 of the D2 subunit (D2-K317). This pathway likely facilitates proton egress during the $S_2 \rightarrow S_3$[25–28] and $S_3 \rightarrow S_0$[10,18,25–27,29–31] transitions. The "O1" or "large" channel leads from $Y_Z$ and the $Mn_4CaO_{5-6}$ cluster's Ca ion past O1. Recent studies have focused on this channel as the primary entry pathway of substrate water[16,17,19–24,28,32]. Asparagine-298 (D1-N298) provides a link between $Y_Z$ (via its proton acceptor, D1-H190) and the water wheel (waters W26, W27, W28, W29, and W30) that has been proposed to serve as a water delivery site via the O1 channel[17,19]; tautomerization to its imidic acid form might support a proton translocating role of D1-N298A[33]. The critical importance of D1-N298 in the water oxidation reactions of PSII has been demonstrated by

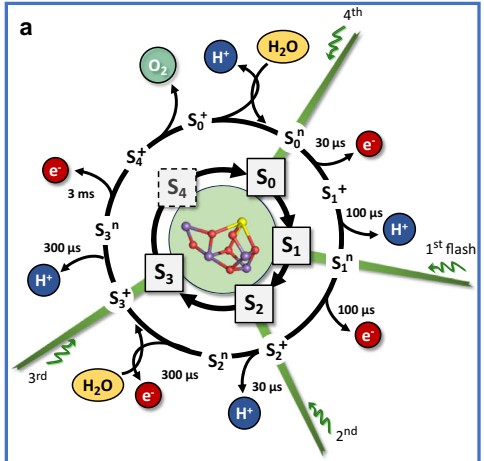
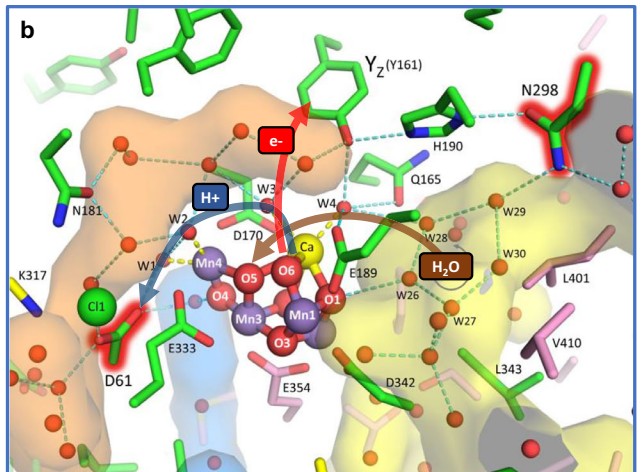

**Fig. 1 | Light-driven reaction cycle of PSII water oxidation and active-site $Mn_4CaO_{5-6}$ cluster with water molecules and selected residues in its environment. a** *Water oxidation cycle:* Starting in the dark-stable $S_1$ state, four consecutive light flashes drive PSII through the five semi-stable S-states of Kok's classical S-state cycle (inner circle of $S_i$-states[6]). By alternating oxidation ($e^-$) and proton removal ($H^+$) steps, the $Mn_4CaO_x$ cluster accumulates up to 4 oxidation equivalents, while avoiding excessive charge accumulation (outer cycle of $S_i^{n/+}$ states[11]). The electrons are removed by transfer to the previously oxidized tyrosine residue ($Y_Z$); the protons move to the periphery of PSII; two water molecules are inserted[3,87]; $O_2$ is formed and released. Approximate room temperature values of the respective time constants are indicated[13]. **b** *Active-site structure:* Water molecules and selected residues in the vicinity of the $Mn_4CaO_{5-6}$ cluster based on the 2.07 Å two-flash ($S_3$-rich) crystallographic model (PDB ID: 6DHO, PSII from *Thermosynechococcus* *vestitus* BP-1; also used for $Mn_4CaO_6$ cluster structure in panel a)[16]. Residues from the D1, D2, and CP43 subunits are depicted in green, yellow, and pink, respectively; Mn in purple, Ca in yellow, Cl in green. Water molecules and six μ-oxo bridges (O1, O2,..., O6) are depicted as red or orange spheres. The pentagon of H-bonded water molecules, consisting of W26 to W30, has been termed "water wheel" and brought into play as an element of substrate water delivery[16,17,19,46]. Numbering of water molecules is based on ref. 16. Approximate boundaries of the Cl1, O4, and O1 channels near the $Mn_4CaO_{5-6}$ cluster are depicted in orange (Cl1), blue (O4), and yellow (O1). The proposed directionalities in proton-coupled electron transfer and water insertion in the oxygen-evolving $S_3 \rightarrow S_4 \rightarrow S_0$ transition are indicated, as are the two key residues investigated here by genetic modification (D1-D61 and D1-N298).

mutagenesis and IR studies. First, mutations of D1-N298 substantially decrease steady-state rates of $O_2$ evolution detected under high-light (saturating) continuous illumination[34,35], but the mechanistic origin of the decreased rate could not be identified. Second, from static FTIR studies it was concluded that the N298A mutation perturbed the network of H-bonds that surround the $Mn_4CaO_{5-6}$ cluster and substantially diminished the efficiencies of the $S_2 \rightarrow S_3$ and $S_3 \rightarrow S_0$ transitions[34]. Third, time resolved FTIR studies show that the N298A mutation decreases the rates of the $S_1 \rightarrow S_2$ and $S_2 \rightarrow S_3$ transitions 3-7-fold, but the oxygen-evolving $S_3 \rightarrow_4 \rightarrow S_0$ transition was not investigated[28]. Here, we show that the N298A PSII variant can undergo the oxygen-evolving $S_3 \rightarrow S_4 \rightarrow S_0$ transition equally fast and efficiently as the wild-type PSII. A paradox— between the severely reduced $O_2$-evolution rate under continuous illumination and the rapid $O_2$ formation detected by time-resolved polarography—is identified and resolved by a scenario supported by time-resolved IR experiments: a severe slowdown of substrate-water insertion *after* $O_2$ formation by the sidechain exchange in the N298A variant.

We also conducted time-resolved IR studies on the D1-D61A variant of PSII, for comparison with wild-type PSII and the N298A variant. Comparison with D61A is especially informative because of its clearly pivotal role in PSII water oxidation. It facilitates $Mn_4CaO_{5-6}$ cluster oxidations[10,36-39] and mutations of D61 substantially decrease steady-state rates of $O_2$ evolution[27,29,40], perturb the H-bond networks surrounding the $Mn_4CaO_{5-6}$ cluster[25,30], and diminish the efficiencies[25,30] as well as rates[29,41,42] of the individual S-state transitions, all this likely relating to its role as an acceptor of protons removed from the substrate water molecules[10,37,43,44]. (In the following the mutations D61A and N298A of the D1 protein subunit will be simply referred to as D61A and N298A).

The present investigation addresses three roles of the protein-water environment of the catalytic metal cluster in the oxygen-evolving $S_3 \rightarrow S_4 \rightarrow S_0$ transition, clearly demonstrating that the H-bond protein-water clusters surrounding the $Mn_4CaO_{4-6}$ cluster are not merely a passive protein-water matrix but also an essential part of the evolutionary fine-tuned active site. (a) Confirming previous experimental studies[29,45], we now find by time-resolved IR spectroscopy in the D61A variant a drastic slowdown of the rate-limiting step of $S_4$-formation, explainable by the crucial role of the D61 in the proton transport path for removing a proton from a Mn-bound substrate hydroxide concomitantly with oxyl radical formation[10]. (b) For the N298A variant, the combination of time-resolved $O_2$-polarography and IR spectroscopy provides evidence that the insertion of a "fresh" substrate water after $O_2$-release is optimized by the protein-water network. We discover that the N298A modification likely causes a pronounced slowdown of the deprotonation-coupled water insertion step, which is also investigated by molecular mechanics (MD) simulations for ca. $10^6$ atoms over a total simulation period of $5\,\mu s$ per PSII variant. (c) Unexpectedly, we encounter a third role of the protein-water network by analysis of the temperature dependence of the $O_2$-formation step. The enthalpic and entropic contribution to the free energy of activation are strongly affected by the exchange of a single residue (N298A), with an almost unvaried rate constant value at room temperature. This interesting case of entropy-enthalpy compensation is also investigated by extensive MD simulations. Finally, we discuss possible implications for water oxidation in inorganic catalyst materials.

## Results
### Targeted protein variations, oxygen-evolving activities, and verification of site mutations
Two PSII point mutants of *Synechocystis* sp. PCC 6803 were investigated for which previous investigation have suggested that they could affect the processes in the oxygen-evolving $S_3 \rightarrow S_0$ transition (see Fig. 1a) in distinctly different ways. A specific aspartate (D61 of the D1 protein) has been frequently suggested to act as a pivotal

**Table 1 | Oxygen-evolution and electron transfer rates for the $S_3 \rightarrow S_0$ transition determined by $O_2$-polarography and infrared spectroscopy**

|  | Wild-type | Asp61→Ala | Asn298→Ala |
|---|---|---|---|
| **Steady-state $O_2$ rate ($\mu$mol (mg of Chl)$^{-1}$ h$^{-1}$)** | | | |
| Cells ($T$ = 25 °C) | 665 ± 35, 680 ± 30[a] | 129 ± 41[a] | 74 ± 7[b] |
| Core complexes ($T$ = 25 °C) | 5010 ± 50[b] | 900 ± 270[b] | 670 ± 170[b] |
| **Time-resolved $O_2$ polarography—reaction time constant (ms)** | | | |
| thylakoids ($T$ = 10 °C) | 3.0 ± 0.3[b], 5.5[c] | 99[c] | 1.9 ± 0.3[b] |
| **Time-resolved IR spectroscopy—reaction time constant (ms)** | | | |
| core complexes ($T$ = 10 °C) | 4.7 ± 0.3[b] | 360 ± 58[b] | 6.5 ± 0.8[b] and 75 ± 9[b] |

The steady-state $O_2$ rates were measured polarographically for illumination with continuous light of saturating intensity. The reaction time constants provide the inverse rate constants of the rate-limiting step in $O_2$-formation. They were determined from time-resolved $O_2$ polarography and infrared spectroscopy using sequences of saturating light flashes for initiating S-state cycle transitions. For further details, including estimation of error ranges, see the "Methods" section and Supplementary Figs. 8–10.
[a]Results from Chu, Nguyen, and Debus[40].
[b]Results from this study.
[c]Results (extrapolated) from Bao and Burnap[45].

proton acceptor in several of the S-state transitions, transiently accepting protons removed from substrate water molecules[25,27,29-31]. Recently, a very specific proposal on its role in the rate-determining step(s) of the oxygen-evolving transition has been presented[10]. Briefly, the rate-determining step is the formation of a reactive oxyl radical, which is facilitated by coupling the transfer of an electron to the previously light-oxidized tyrosine sidechain ($Y_Z$ in Fig. 1b) to PT from a metal-bound hydroxide to D61, the latter involving three further metal-bound water species. This $S_3 \rightarrow S_4$ transition is preceded by deprotonation of a glutamate pair (D1-E65−D2-E312, not shown in Fig. 1b) in the $S_3^+ \rightarrow S_3^N$ transition that precedes the proton-coupled electron transfer and likely "prepares" the D61 by structurally mediated increase in its proton affinity[10,44], in line with recent results obtained by time-resolved crystallography[46]. Thus, the exchange of the protonatable carboxylate sidechain of the D61 against an inert methyl group (D61A mutant) is predicted to profoundly affect light-driven oxygen evolution[10,37,44,47]. Indeed, a strong decrease in the rate of oxygen evolution is observed for the D61A variant (Table 1), in good quantitative agreement with results previously obtained for the same genetic variant[31,45].

The second here selected protein variation is the asparagine-alanine exchange at position 298 of the D1-protein (N298A mutant). It is more remote from the site of oxyl radical formation and O-O bond formation, and thus it is not expected to be directly involved in these reactions. An indirect, possibly energetic effect on these processes might result from H-bonding of the asparagine to the histidine (H190) that accepts a proton upon $Y_Z$ oxidation and shuttles it back in the $S_3 \rightarrow S_4$ transition. Yet of particular interest regarding the role of this residue is its association with the "water wheel" comprising W29, W28, W26, W27, and W30, which has been suggested, based on crystallographic data[16,17,19,46], to be crucial in water delivery to the active-site metal cluster.

Even though the N298 residue likely is not related to the rate-determining step of oxyl radical formation in the $S_3 \rightarrow S_4$ transition, the light-saturated steady-state rates of $O_2$ evolution of the N298A variant were at least equally strongly diminished as found for the D61A residue exchange, see Table 1. The reduction in the $O_2$ evolution in comparison to unmodified wild-type (WT) protein is similar to those reported previously in *Synechocystis* sp. PCC 6803[34] and *Chlamydomonas reinhardtii*[35] for the N298A and D61A variants.

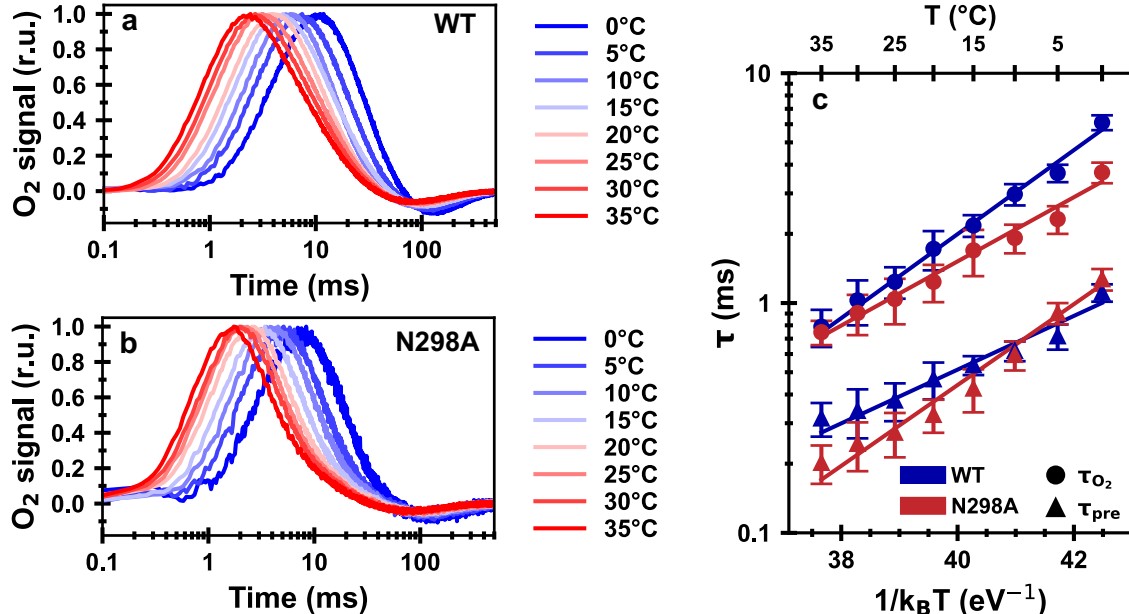

**Fig. 2 | Time-resolved O$_2$-evolution polarography on PSII thylakoids from Synechocystis sp. PCC 6803.** Normalized O$_2$-evolution transients of **a** wild-type and **b** N298A PSII thylakoids at eight different temperatures (0–35 °C). **c** Arrhenius plot of time constants extracted from (**a**, **b**) using a diffusion model. The time constant $\tau_{O2}$ reflects the oxygen evolution rate ($\tau_{O2} = 1/k_{O2}$); $\tau_{pre}$ corresponds to proton removal preceding O$_2$ evolution. The slope of the Arrhenius plot reveals activation energies for the preceding reactions of 269 ± 25 meV (WT) and 405 ± 24 meV (N298A), and for the oxygen-evolution step of 421 ± 22 meV (WT) and 321 ± 18 meV (N298A). For the oxygen evolution step, the Arrhenius parameters and the corresponding enthalpic and entropic contributions to the free energy of

activation are shown in Table 2, for the preceding reaction in Supplementary Table 1. An overview of all fit parameters involved in simulation of the O$_2$-evolution transients is provided in Supplementary Fig. 6. At each temperature, at least three independent data sets were obtained and averaged, before determining the time constants by simulation and fit with the diffusion model described in ref. 29; the error bars in panel **c** correspond to parameter uncertainties derived from the covariance matrix (calculated by the least-squares minimization algorithm). For a direct comparison of O$_2$-evolution transients of wild-type and N298A at 20 °C, see Supplementary Fig. 5.

**Table 2 | Results from applying Arrhenius and Eyring analysis to the O$_2$-polarography data corresponding to the oxygen evolution step of wild-type PSII and PSII carrying the N298A mutation**

| Synechocystis sp. PCC 6803 variant | $\tau_{O2}$ at 20 °C (ms) | ln($A$/s) | $E_a$ (meV) | $\Delta H^{\ddagger}$ (meV) | $-T_o\Delta S^{\ddagger}$ (meV) | $\Delta G^{\ddagger}$ (meV) |
|---|---|---|---|---|---|---|
| Wild-type | 1.72 ± 0.34 | 23.0 ± 1.3 | 421 ± 22 | 396 ± 22 | 187 ± 28 | 583 ± 6 |
| N298A | 1.24 ± 0.23 | 19.3 ± 1.1 | 321 ± 18 | 296 ± 18 | 280 ± 24 | 575 ± 6 |

The natural logarithm of the pre-factor, $A$, as well as the activation energy, $E_a$, were obtained from the Arrhenius plot in Fig. 2c. From those values, the enthalpy ($\Delta H^{\ddagger}$), entropy ($-T_o\Delta S^{\ddagger}$) and Gibbs free energy of activation ($\Delta G^{\ddagger}$) were calculated using Equation S1, with $T_0$ = 20 °C. The time constant of oxygen evolution at 20 °C ($\tau_{O2}$) is also shown.

In the D61A variant, the decreases in steady-state oxygen evolution corresponds to a reaction time constant that is increased by clearly more than one order of magnitude. (Note that the influence of the sidechain exchange on the time-resolved O$_2$ polarography data is expected to be greater than that on the corresponding steady-state rate; this is expected because the latter is determined by a co-limitation of acceptor and donor side processes). In the N298A variant, however, the severely reduced steady-state rate of oxygen evolution is not matched by a deceleration of O$_2$-evolution time constant, but rather a slight acceleration from 3.0 to 1.9 ms is observed for O$_2$-evolution at 10 °C. To explain this divergence, we propose that in the N298A variant the reaction steps of oxyl radical formation, O-O bond formation and O$_2$ release are largely unaffected by the residue exchange, whereas subsequent steps of insertion of a hydroxide in the void left by O$_2$-release are slowed down significantly.

Recently the posttranslational modification or even restoration of genetic variations obtained by site-directed at the donor side of PSII has been described[48]. To verify the presence of the D1-A61 or D1-A298 residues in mutation-bearing PSII core complexes[48,49], the amino acid sequences of chymotryptic or chymotryptic/tryptic peptides of isolated D1 subunits were analyzed by mass spectrometry. In N298A, at

least 99% of the residue at the mutation position was the desired alanine (see Supplementary Figs. 1 and 2). In D61A, approx. 98.5% of the residue at the mutation position was the desired alanine (see Supplementary Figs. 3 and 4). These results are in marked contrast to the post-translational restoration of the wild-type amino acid residue recently observed in the mutants D1-D170H[48,49], D1-E189Q[48], and D1-D342N[48].

**Enthalpy and entropy of activation from time-resolved O2-polarography**

Oxygen release transients were measured at temperatures ranging from 0 to 35 °C, for wild-type and N298A thylakoid membranes (Fig. 2a, b). The O$_2$ transients were simulated with a diffusion model, as described in the Methods section, to obtain a time constant for O$_2$ evolution ($\tau_{O2}$) and a time constant for a preceding reaction step ($\tau_{pre}$)[29,45]. These time constants likely are assignable to proton removal in the $Y_Z^{ox}S_3^+ \rightarrow Y_Z^{ox}S_3^n$ transition ($\tau_{pre}$) and the overall rate-limiting oxyl-radical formation of the $Y_Z^{ox}S_3^n \rightarrow Y_ZS_4$ transition ($\tau_{O2}$)[13,50,51]. The Arrhenius activation-energy analysis was applied to both sets of time constants (Fig. 2c). Here, the rate constant of oxygen evolution is of special interest. Applying Eyring's transition state theory[52–54], the enthalpic and entropic contribution to the free energy of activation were calculated for the rate constant of

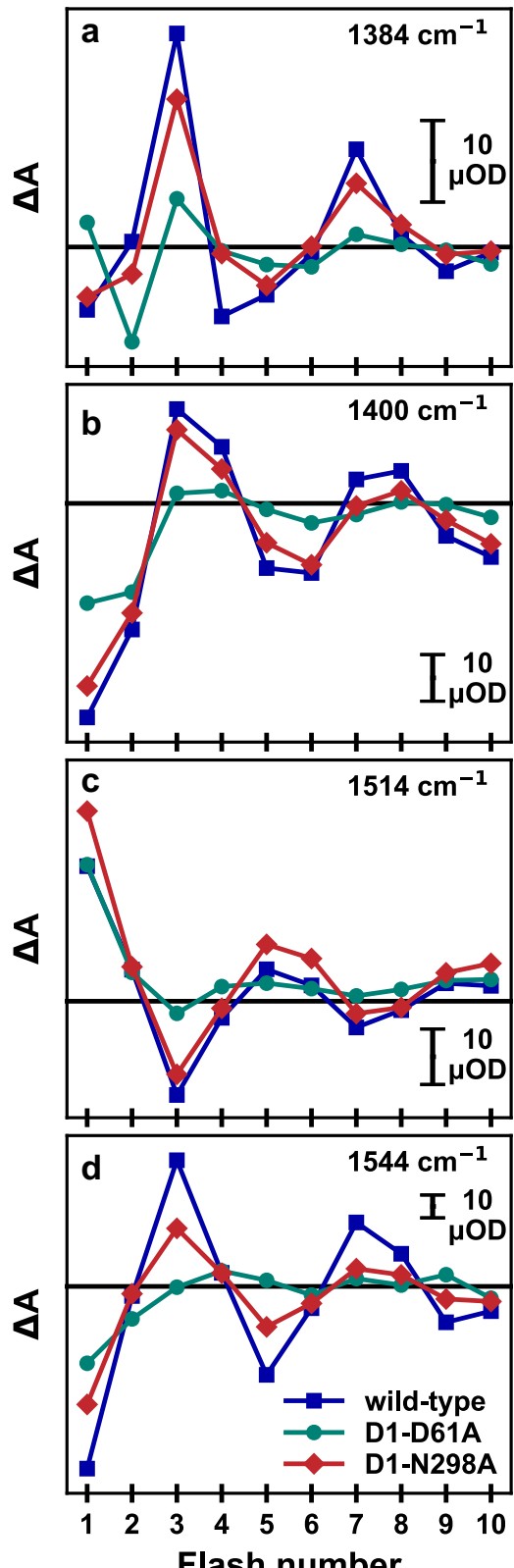

**Fig. 3 | Flash-number dependence of the IR difference absorption of PSII core complexes from Synechocystis sp. PCC 6803.** Data of wild-type PSII (*blue*), as well as PSII with point mutations D61A (*green*) and N298A (*red*) of the D1 protein subunit is shown for **a** 1384 cm⁻¹, **b** 1400 cm⁻¹, **c** 1514 cm⁻¹, and **d** 1544 cm⁻¹. Each data point was obtained by averaging the IR difference absorption between 700 and 800 ms following the *n*th excitation flash (*n* = 1,..., 10). The exciting laser flashes were spaced by 1 s.

oxygen evolution ($k_{O2} = 1/\tau_{O2}$) as described previously[10,45]; the applicability of transition state theory is discussed elsewhere[55]. The remarkable result from this analysis is that the N298A residue exchange increases the entropy of activation significantly, but also decreases the activation enthalpy such that the rate constants in the room temperature range remain similar to that of the wild-type PSII.

### Static FTIR difference spectra

The static FTIR S-state difference spectra of Supplementary Fig. 7 inform on changes in the protein-water environment of the Mn₄Ca-oxo cluster that are associated with the respective S-state transition and detectable after its completion. Minor spectral changes are detectable in the mid-frequency $S_2$-minus-$S_1$ spectrum (Supplementary Fig. 7a), which resemble alterations produced by the mutations of many other residues in the protein-water environment near the Mn₄Ca-oxo cluster (see Table 1 in ref. 56). This suggests that the N298A mutation also perturbs the protein-water environment that couples to the S-state transitions, as was concluded previously[34] and discussed for D61A elsewhere[30]. Perturbation of the protein-water environment in both PSII variants, N298A and D61A, is shown more directly in the weakly H-bonded O-H stretching region of the IR spectra (Supplementary Fig. 7b).

Comparison of the static difference spectra does not provide evidence for major modification of the S-state transitions in either N298A or D61A. The extent of spectral modifications in N298A is especially minor, also in the structure-sensitive amide I and amide II regions, rendering conformational changes of the protein environment in the vicinity of the Mn₄Ca-oxo cluster unlikely.

### Time-resolved IR reveals biphasicity in the O2-evolution transition of D1-N298A

PSII was driven through its reaction cycle with its four semi-stable S-states (Fig. 1a) by application of a sequence of 10 laser flashes (532 nm, ca. 5 ns halfwidth) with a 1 s interval between flashes. Flash-induced IR transients were obtained at selected wavenumbers for wild-type, N298A, and D61A; see Supplementary Figs. 13 and 14 for flash-induced transients of all four S-state transitions. To reach a noise level below 5 μOD, signal averaging of data collected for 500–5000 flash sequences applied to dark-adapted PSII core complexes was required at each wavenumber selected for investigation (Supplementary Table 2). The tuneable quantum cascade laser (QCL) laser system was able to cover the spectral range from 1650 to 1300 cm⁻¹, but to match the noise level requirements with the available amount of PSII core complexes, gapless coverage of this spectral range was out of the question and a selection of a small number of informative wavenumbers was approached.

The wavenumbers of 1384, 1400, 1514, and 1544 cm⁻¹ were selected for the following reasons: Based on the FTIR difference spectra of the $S_3 \rightarrow S_0$ transition of Supplementary Fig. 7 and results described in ref. 10, at 1384, 1514 and 1544 cm⁻¹, comparably large contributions of the millisecond phase of oxygen evolution were expected (and indeed found); the 1400 cm⁻¹ wavenumber was chosen to facilitate direct comparison to results obtained in refs. 28,57,58. The 1384 cm⁻¹ and the 1400 cm⁻¹ wavenumbers are predominantly associated with the stretching vibrations of carboxylate residues ligating the metal ions of the Mn₄Ca-oxo cluster[59–61]. At 1514 cm⁻¹ a strong contribution of the CO stretching vibrations of the redox-active tyrosine radical has been predicted[62] and therefore this wavenumber was selected. However, we find that the IR transients at 1514 cm⁻¹ do not match the expected oxidation and reduction kinetics of $Y_Z^{ox}$ in the various S-state transitions suggesting that prominent signals of other origin(s) overlap. At 1544 cm⁻¹ the amide II vibrations become visible[61,63], which may inform on conformational changes of the protein backbone[63]. Moreover, we collected IR

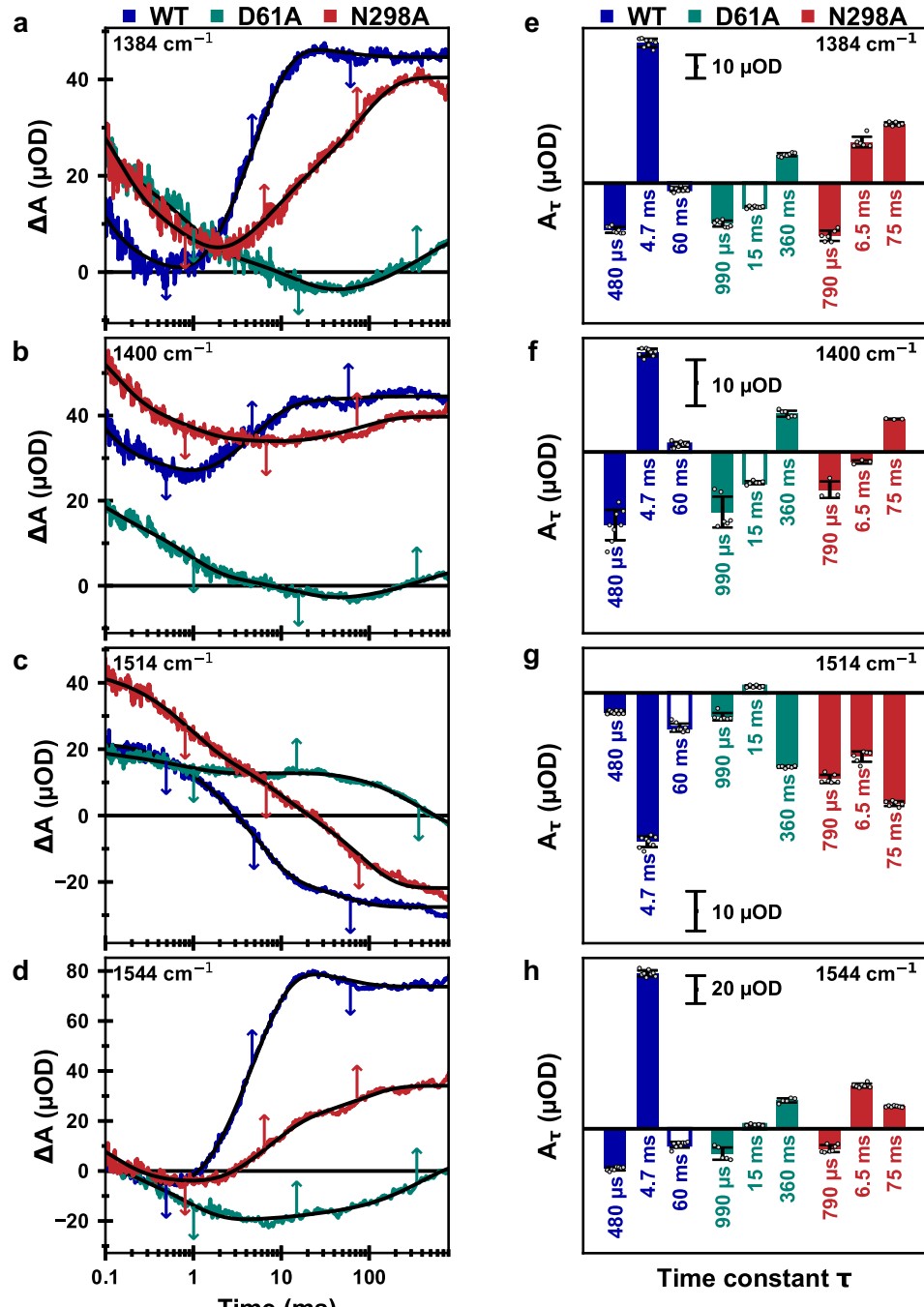

**Fig. 4 | Time-resolved IR data of the $S_3 \rightarrow S_0$ transition of PSII core particles from Synechocystis sp. PCC 6803 at 10 °C. a–d** A deconvolution algorithm was applied to the wild-type and N298A transients of 10 consecutive excitation flashes, resulting in four transients corresponding to the "pure" S-state transitions. These are similar to the transients recorded after the 3rd saturating excitation flash (compare Supplementary Fig. 12). For the D61A variant, the original 3rd flash data is shown without deconvolution, because of uncertainties regarding the likely highly S-state dependent miss factor. Wild-type data is shown in blue, D61A in green, and N298A in red; the black lines were obtained by curve-fitting with the parameters shown in Supplementary Table 4. **e–h** Amplitudes corresponding to the time constants obtained from fitting the data on the left to a sum of exponentials. The bars filled in white are assigned to processes that are not specific for the oxygen-evolving $S_3 \rightarrow S_0$ transitions, see Supplementary Figs. 15 and 16. The time constants were determined globally by a joint fit of transients collected at several wavenumbers with the same set of time constants. Uncertainty ranges were estimated by performing fits of various subsets of the four transients, for which the mean values of the time constants and amplitude are shown here, with the error bars indicating the respective standard deviations of the amplitudes (Supplementary Figs. 8–10 and Tables 3, 4). The time constant values are indicated by arrows in (**a–d**). The estimated uncertainty ranges of the time constants are $4.7 \pm 0.3$ ms and $60 \pm 35$ ms (wild-type), $15 \pm 8$ ms and $357 \pm 58$ ms (D61A), $6.5 \pm 0.8$ ms and $75 \pm 9$ ms (N298A). Further details on the S-state deconvolution, curve-fitting, and estimation of average time constants and their uncertainty ranges are presented in the Methods section and in Section 4 of the Supplementary Materials. For a closer look at the millisecond kinetics at 1384 and 1544 cm$^{-1}$, see also Supplementary Fig. 18.

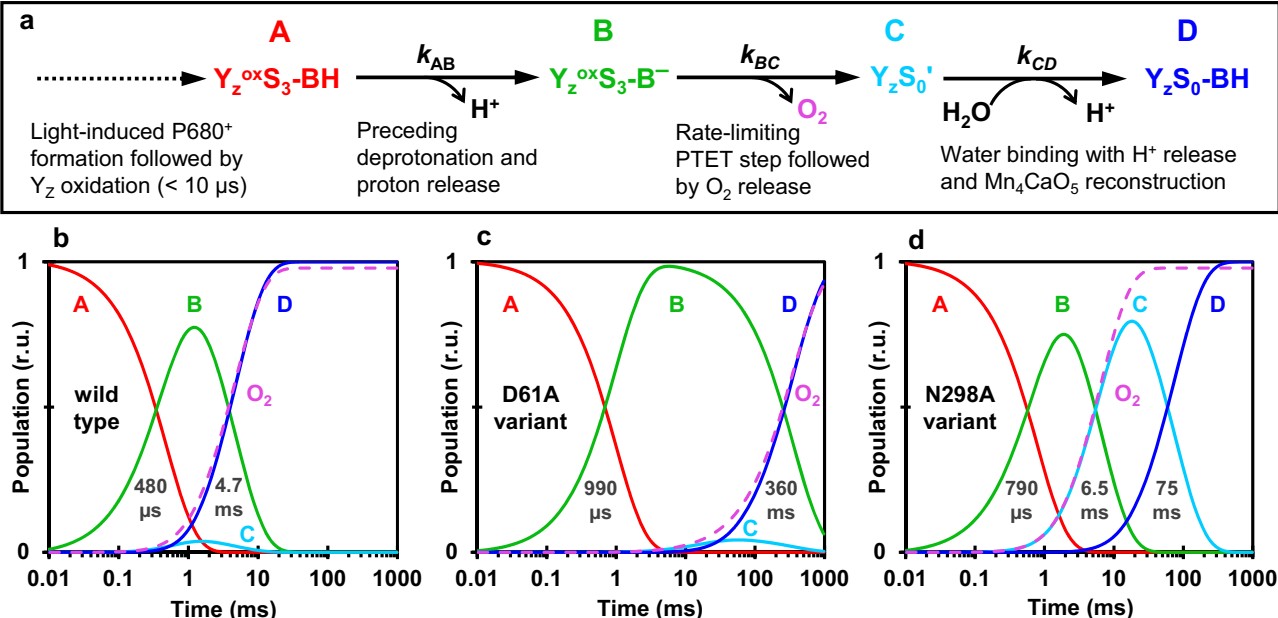

**Fig. 5 | Population of reaction intermediates during the oxygen-evolution transition ($S_3 \rightarrow S_0$).** The populations were calculated based on the reaction scheme of (**a**) and the rate constants experimentally determined in the time-resolved IR experiment ($k = 1/\tau$) for **b** WT PSII, **c** the D61A variant, and **d** the N298A variant; the respective IR time constants are indicated in (**b**–**d**). Each reaction step is assumed to be associated with a significant free energy change so that the backreaction can be neglected. For the WT and the D61A variant, the C intermediate likely does not accumulate to a resolvable extent because $k_{CD}$ is significantly greater (faster) than $k_{BC}$; therefore, $k_{CD}$ values were arbitrarily chosen to equal 20 times the respective $k_{BC}$ values. The dashed violet lines indicate the time courses of produced dioxygen.

transients at 1478 cm$^{-1}$ (Supplementary Fig. 17), which reflect the oxidation state changes of the quinones at the acceptor side of PSII.

Figure 3 shows that the IR difference absorption at 700-800 ms after the excitation flash exhibits a clear flash-number dependence in all data sets. Wild-type PSII and N298A show similar period-of-four oscillating patterns, indicating that both samples are going through the S-state cycle with similar turn-over efficiency. Also, the absolute ΔA values are mostly similar, implying an undiminished amount of O$_2$-evolving PSII in N298A. In D61A, the oscillating pattern is significantly more dampened, but still clearly discernible, indicating that the sample is cycling with a greatly reduced transition efficiency.

In the following, we analyze the IR transients of the oxygen-evolution transition, $S_3 \rightarrow S_4 \rightarrow S_0$, induced predominantly by the third flash applied to dark-adapted PSII (for data on other S-state transitions see Supplementary Fig. 11). The IR transients of the $S_3 \rightarrow S_0$ transition of wild-type PSII show a well-resolved kinetic phase with a time constant of 4.7 ms at all four probed wavenumbers (Fig. 4a–d, *blue*), which concurs with the O$_2$ evolution rate determined by O$_2$ polarography at 10 °C (see Table 1). In the D61A variant this phase is strongly retarded in the IR data (360 ms phase, Fig. 4a–d, *green*), suggesting that the O$_2$-evolution rate is strongly affected by the mutation. The N298A variant, however, displays a more complex behavior. In first experiments, we collected data at 1400 cm$^{-1}$, which at first glance suggested that the O$_2$ evolution in the N298A variant is significantly slowed (Fig. 4b). This, however, would be at odds with the O$_2$-polarography results. Further data sets at 1384, 1514, and 1544 cm$^{-1}$ then revealed biphasic behavior with phases of 6.5 and 75 ms. The amplitudes of these two phases show a clear wavenumber-dependent behavior (Fig. 4e–h); at 1384 cm$^{-1}$ and 1514 cm$^{-1}$ the slower phase dominates, while at 1544 cm$^{-1}$, the faster phase is more prominent; at 1400 cm$^{-1}$ the faster phase is essentially absent. A 60 ms phase is well resolved in the WT and likely assignable to acceptor side processes, whereas the origin of the 15 ms phase in D61A remains unclear; both phases are not specific to the $S_3 \rightarrow S_4 \rightarrow S_0$ transition; see Supplementary Figs. 15 and 16 and their discussion in the Supplementary Information. Simulation of the transients more-over reveal time constants of 480, 790, and 990 µs with the similar

negative amplitudes for the WT, D61A, and N298A variants, which are likely assignable to the PT preceding oxygen evolution[12,50,51,58] and also visible as $\tau_{pre}$ in the O$_2$-transients (Fig. 2). Noteworthily, the PT phase is largely unaffected by both residue exchanges, aside from a moderate slowing down (1.6–2.1-fold), which previously has been seen for the D61A variant also in time-resolved O$_2$-polarography experiments[45].

## Population of intermediate states in the $S_3 \rightarrow S_4 \rightarrow S_0$ transition—a possible scenario

The time-resolved IR experiments suggest that in the $S_3 \rightarrow S_4 \rightarrow S_0$ transition, instead of a single millisecond phase there are two kinetic phases—and thus two processes—resolvable in IR transients of the N298A variant. Since only the faster process coincides with appearance of O$_2$ as detected by time-resolved O$_2$-polarography, the slower time constant relates to a process in the oxygen-evolving transition that occurs after O$_2$-release. In the WT-PSII, it is faster than the O$_2$-evolution step and thus not resolvable in time-resolved experiments. In the N298A variant, however, this process is slowed down by minimally one order of magnitude and thereby becomes detectable as a separate kinetic phase. The existence of this slow process in the N298A variant also resolves the incongruity between fast O$_2$-formation detected by time-resolved polarography and severely reduced rates of O$_2$-evolution detected for continuous illumination at high photon fluence rates (Table 1). Since the slow process in the N298A variant occurs *after* O$_2$-release, we assign it to the water deprotonation and hydroxide insertion needed to refill the vacant O$_5$ site, finally resulting in Mn$_4$CaO$_5$ cluster in its S$_0$ state. Which of these steps, water deprotonation and insertion, represents the rate-limiting bottleneck remains open as does their sequence; likely these processes proceed in a tightly coupled way.

The relation of the experimentally determined rate constants to the populations of the hypothetical intermediates is shown in Fig. 5 for the WT, D61A and N298A variant; the proposed assignment of the B→C and C→D transitions to atomistic processes is schematically shown in Fig. 6.

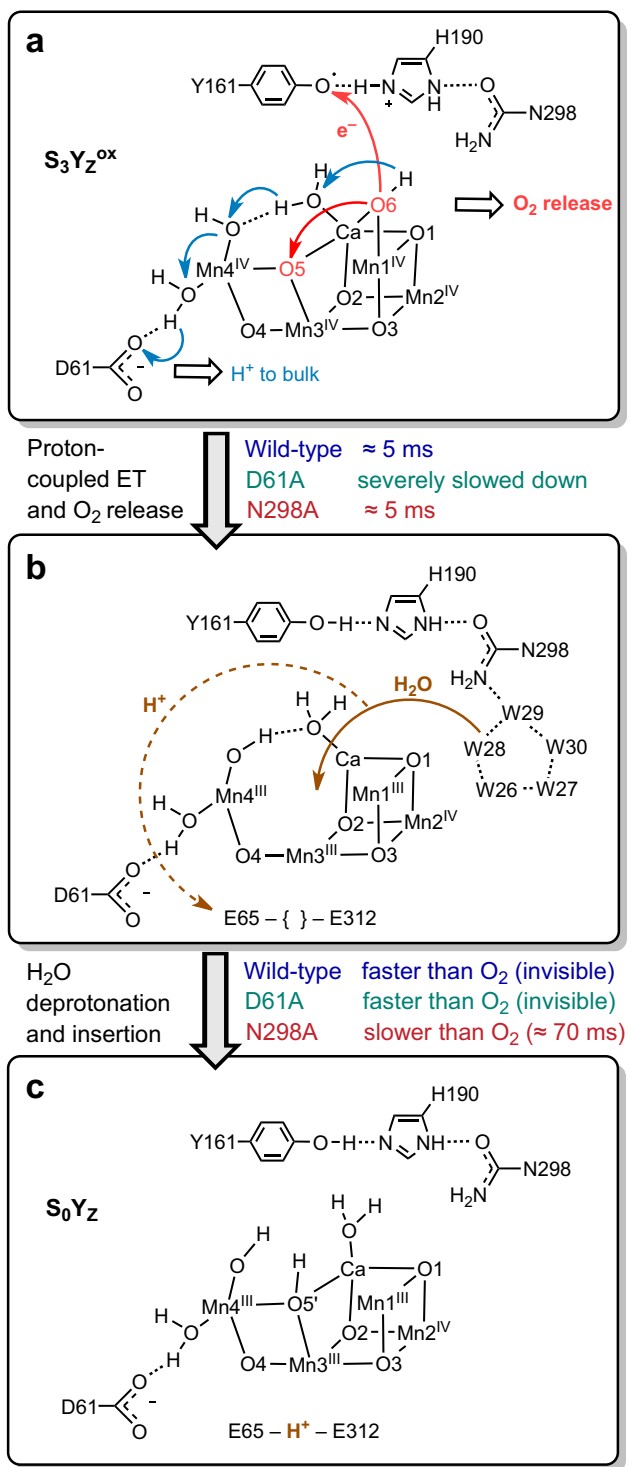

**Fig. 6 | Reaction scheme showing the $Mn_4CaO_x$ cluster as well as the D61, Y161 ($Y_Z$), H190 and N298 residues during the $S_3 \rightarrow S_0$ transition. a** Oxygen atom O6 is oxidized by transferring an electron to $Y_Z$; the proton bound to O6 is concomitantly transferred to D61 in a Grotthus-type mechanism. Subsequently, O6 and O5 form an O=O bond while reducing Mn4, Mn3, and Mn1 from +IV to +III. This step takes place within about 5 ms in wild-type PSII as well as in PSII containing the N298A mutation, but is severely slowed down in the D61A mutant. Here a specific mode of O-O bond formation is indicated[10,43,88,89]; alternatives have been discussed[1,90]. **b** A new water molecule is inserted into the $Mn_4CaO_x$ cluster, and a proton is simultaneously released to the bulk, leading to the formation of the $S_0$ state (shown in **c**). This step is faster than $O_2$ release and thus "invisible" in wild-type PSII as well as in PSII containing the D61A mutation. We propose that in the N298A mutant, this step is slowed down roughly ten-fold and thus becomes visible for spectroscopic methods. The given time constants are approximate values at 10 °C. The arrangements of the non-hydrogen atoms schematically shown in (**a**, **c**) correspond to crystallographically resolved structures[16,46]. The structures in panel b, the location of H-atoms, and all the particle movements indicated by arrows have been deduced by structure-based computational chemistry previously[10] and thus are more hypothetical than the arrangement of non-hydrogen atoms shown in (**a**, **c**).

variant. Due to the homodimeric organization of PSII, this corresponds to a total of 10 trajectories of 500 ns duration. Neglecting the first 100 ns of each trajectory, a total simulation time of 4 μs was analyzed, with selected results shown in Figs. 7 and 8. The overall structural stability of the simulated systems during the unrestrained MD simulations was proven by calculating the time evolution of different properties (see Supplementary Fig. 19).

Focusing on the water molecules shown in the MD snapshots of Fig. 7a, the MD simulations reproduce most of the various water positions observed by cryo-EM[64]. These results are also consistent with those obtained for *T. vestitus* (see Supplementary Fig. 20). The latter are compared to X-ray crystallography[46] and fit especially well with a recent cryo-EM structure[24], which has added several transiently or partially occupied water positions (W4a, W4b, W26a). The main effect of the N298A modification may be described as a less constrained conformational space in the vicinity of this residue, corresponding to about 20% increase in local volume compared to the wild-type (Fig. 7b). By exchange of the asparagine sidechain against the smaller alanine sidechain, the water wheel (W26−W30) gains space and expands into the gained space (Fig. 7a), which is not coupled to an increase in the average number of contributing water molecules but rather associated with a moderate decrease from about 5.0 down to 4.5 (Fig. 7c and Supplementary Table 5). Thus, the wild-type water wheel may be seen as more constrained when compared to the N298A variant, possibly associated with a higher free energy of the water-wheel structure and thereby favoring the water insertion step in the wild-type PSII by a reduced free energy difference between the initial state and the transition state. This aligns with our finding that water molecules are nearly three times less likely to transition from the water wheel to the $Mn_4$Ca-oxo cluster in the N298A mutant (Fig. 7d).

A sizeable entropic contribution to the free energy of activation of the rate-determining step of the oxygen-evolution transition ($-T_0\Delta S^{\ddagger}$ in Table 2) has been assigned to a high number of conformations of the H-bonded protein-water network in the surrounding of the $Mn_4$Ca-oxo cluster and $Y_Z$, with only a specific subset of these acting as an active conformation that allows the rate-limiting step to proceed[10,55]. Our MD simulations illustrate and support this entropic contribution by showing pronounced fluctuations (see also Supplementary Movie 1). These include H-bond distances and protein conformations that relate directly to the reaction coordinate(s) and the likely transition state structure, as shown in Fig. 8 and summarized as follows:

1. The H-bond to the second nitrogen ($N_\delta$) of His190 to the asparagine sidechain in the WT is substituted in the N298A

### Molecular dynamics simulations rationalize the influence of the asparagine-alanine exchange

MD simulations were performed on the cryo-EM structure from the mesophilic cyanobacterium, *Synechocystis* sp. PCC 6803[64]. To model the $S_3Y_Z^{ox}$ state, the coordinates of $Mn_4CaO_6$ cluster and the C-terminus of D1 (D1-Ala 344) were taken from the crystallographically characterized PSII of *Thermosynechococcus vestitus* BP-1[46]. Force-field parameters for the MnCa-oxo cluster were derived as described in the Supplementary Information. For the complete PSII dimer model immersed in a lipid-bilayer–water environment (about $10^6$ atoms), we calculated five trajectories of 500 ns for wild-type PSII and the N298A

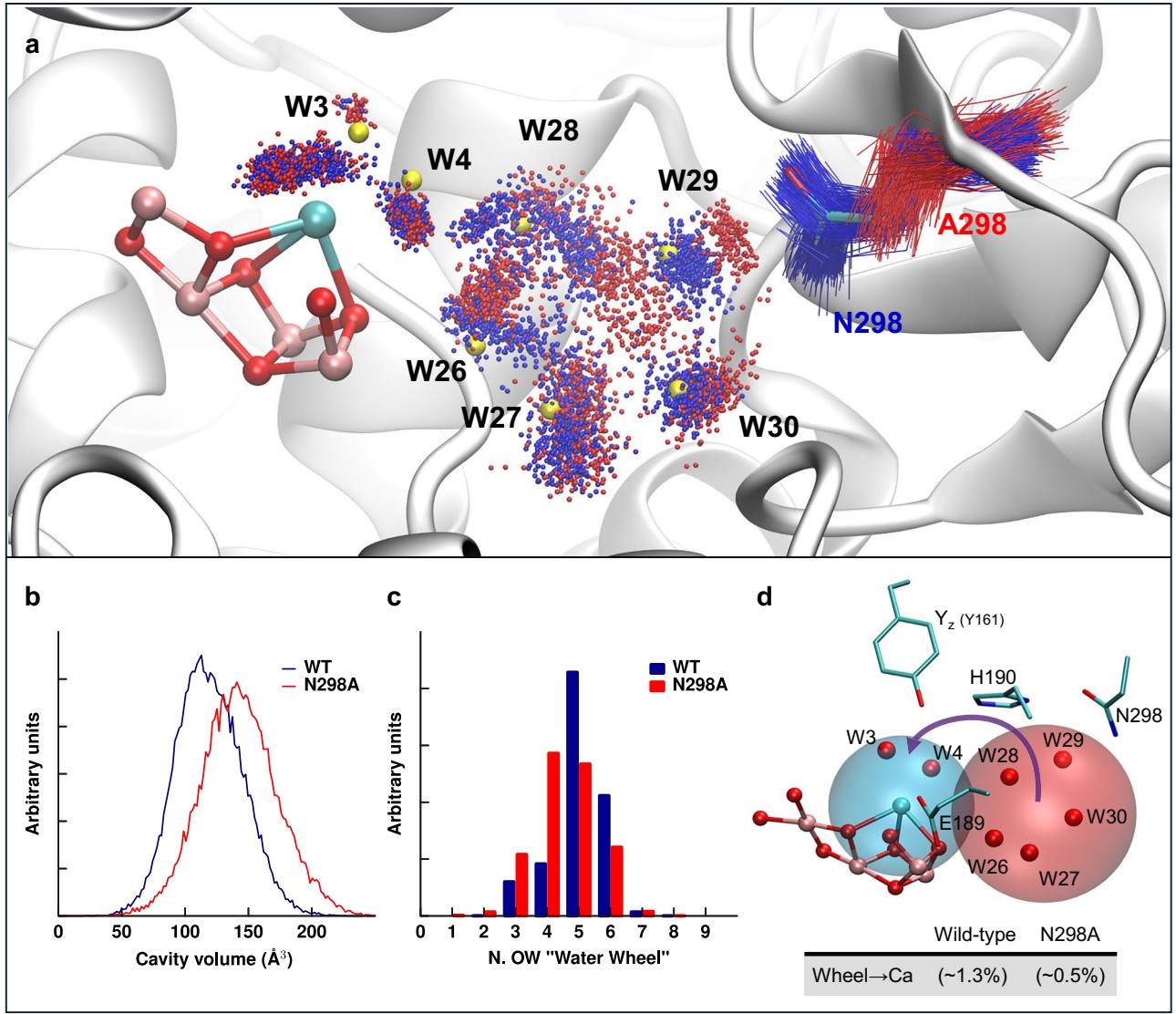

**Fig. 7 | Spatial boundaries, occupancy, and water exchange in the water wheel region. a** Water distributions and MD conformations around residue N298 (wild-type, blue) and A298 (N298A mutant, red) are represented by extracting 40 snapshots by each of the 10 independent 400 ns-long trajectories. Water wheel molecules were selected within a 3.8 Å-radius sphere centered on the center of mass of the $C_\alpha$ atoms of residues V185, H190, L343, A344, CP43-A386, and CP43-I398. Red and blue dots and lines represent water molecules and residues. A less frequently sampled cluster in the upper left identifies a third, alternative water binding site on the calcium, consistent with recent structural data for *T. vestitus*[24] (see Supplementary Fig. 20). Yellow spheres indicate the five waters of the penta-gon, and W3 and W4 from a cryo-EM structure (PDB ID: 7N8O, *Synechocystis sp*. PCC 6803)[64]. MD snapshots were fitted to this structure using the backbone atoms of D1-protein residues 165–210, 303–344, and CP43-protein residues 366–418. **b** The cavity boundaries for wild-type (blue) and N398A (red) were defined using a central 6 Å sphere centered on the water wheel, and an outer 4.5 Å-radius sphere centered on the $Mn_4Ca$ cluster; see Supplementary Information for details. **c** Histograms of the water occupancy for the "water wheel" within the 3.8 Å-radius spherical region (detailed in panel a) for wild-type (blue) and N298A (red). **d** The $Mn_4CaO_{5-6}$ cluster and its surrounding environment, including waters W3, W4 (calcium-bound), W26–W30 (water wheel), and selected residues. We analyzed the water exchange between the cyan 3.0 Å-radius sphere centered on the Ca and the red 3.8 Å-radius sphere centered on the water wheel cluster. Table reports the probability of water transitions from the red to the cyan sphere, estimated from transition events within consecutive 10 ps intervals.

variant by rarer (11% probability) H-bonding to the backbone carbonyl of L297 in competition with water molecules (Supplementary Figs. 21 and 22), thereby likely increasing the conformational variability of the H190 sidechain (Supplementary Fig. 23).

2. For the $Y_Z$-H190 pair, the MD simulations do not reveal changes in H-bonding itself (Supplementary Fig. 24) but rather highlight pronounced fluctuations in the relative orientation of the planes formed by their rings (Fig. 8e). We note that the relative $Y_Z$–H190 position changes upon $Y_Z$, oxidation/reduction, as observed experimentally in crystallographic snapshot data[23,46] and computationally[10], indicating a close relation to the respective reaction coordinate. The $Y_Z$–H190 motions are influenced by the two conformers associated with the second nitrogen of His190 (Fig. 8a and Supplementary Fig. 25), which also appear to modulate the number of water molecules populating the water wheel as defined in Fig. 7, possibly perturbing its structure and dynamics (Supplementary Fig. 26).

3. The MD simulations reveal a pronounced influence on the position of the Glu189 sidechain (Fig. 8f, g). Here, the active con-formation in oxyl radical and subsequent O-O bond formation likely corresponds to the occupancy peak of the WT at 3.5 Å[55]; this occupancy peak is not only strongly diminished but also highly "blurred" in N298A, resulting a broad plateau level, which indicates a severe destabilization of the active conformation compared to WT PSII.

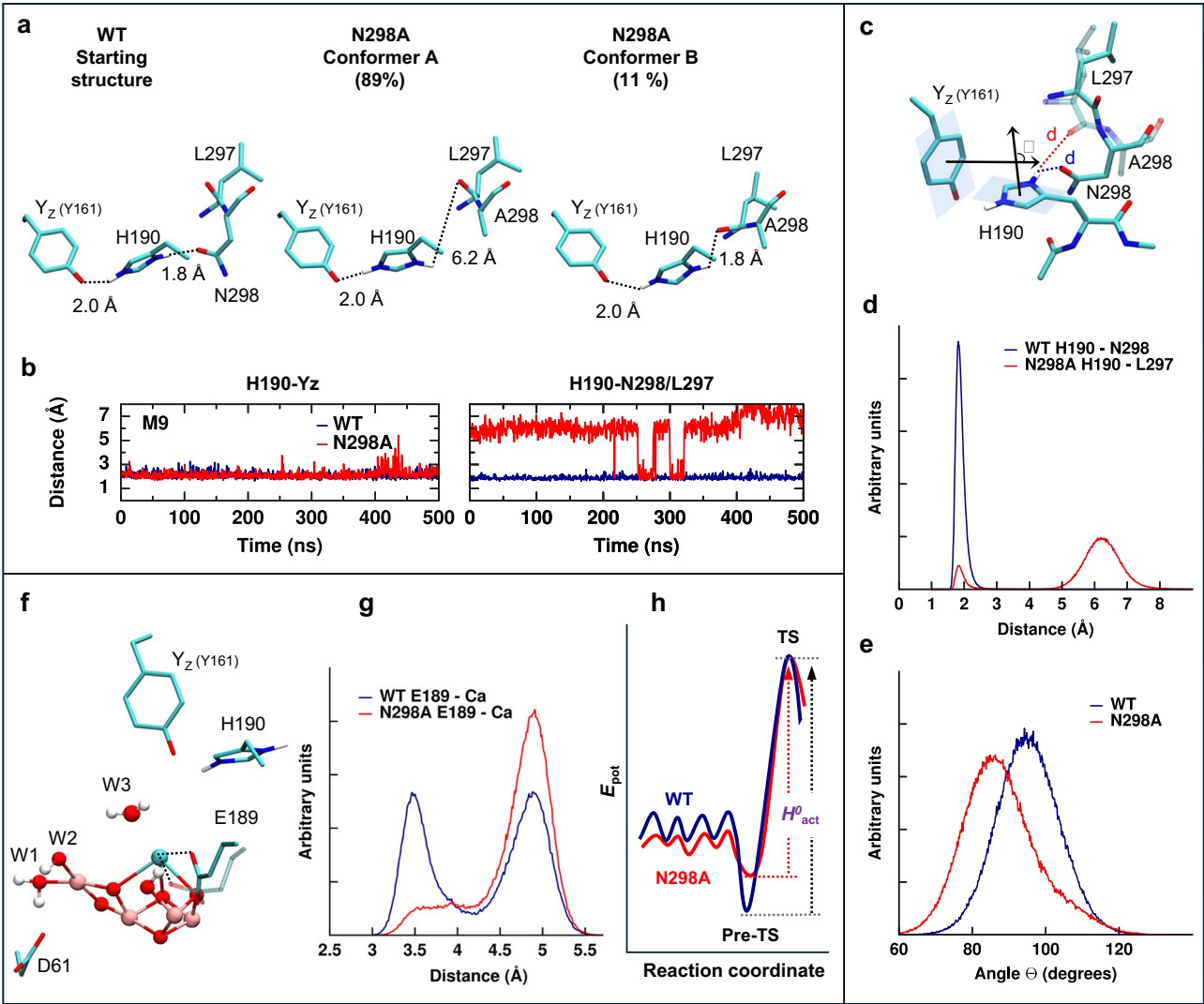

**Fig. 8 | Conformational dynamics linked to enthalpy-entropy compensation.**
**a** Wild-type (WT) PSII simulations show a strong H-bond of the $Y_Z$–H190–N298 triad. In the N298A simulation, the backbone carbonyl of L297 partly substitutes the asparagine in the H-bond network. **b** Fluctuations of the H-bonds involving the epsilon nitrogen ($N_\epsilon$) and delta nitrogen ($N_\delta$) of H190 in one representative monomer trajectory (i.e., M9) for both the wild-type (blue) and the N298A mutant (red); more details are reported in Supplementary Fig. 21. **c** Representation of key geometrical features around H190: the angle $\Theta$ between the planes formed by the rings of $Y_Z$ and H190 and the distance $d$ of the H-bond of the ($N_\delta$) nitrogen.

**d** Distribution of distances between the hydrogen attached to the delta nitrogen ($N_\delta$) of H190 and N298 (wild-type, blue) or L297 (N298A mutant, red). **e** Distribution of the angle, $\Theta$, between $Y_z$ and H190 in wild-type (blue) and the N298A mutant (red), reflecting the relative orientation of planes defined in panel c. **f** Key residues likely involved in the oxyl-radical formation step. **g** Distribution of E189-Ca distances. **h** Illustration of a hypothetical energy landscape for wild-type (blue) and N298A mutant (red), including a pre-transition state (Pre-TS) and transition state (TS). The softer profile in the N298A mutant may explain an increased entropy and decreased enthalpy of activation ($H^0_{act}$).

Currently, it is not possible to quantify the entropic and enthalpic contribution to $\Delta G^\ddagger$ of the rate-determining step, but the results of our MD simulations render an increased entropic and concomitantly decreased enthalpic contribution plausible. Denoting the active conformation of the protein-water network reached before the onset of the rate-limiting step in $O_2$ formation as the pre-transition state (pre-TS), a softer energy landscape for the N298A mutant may result in a less stabilized pre-TS, thereby increasing the effective entropy of activation and decreasing the effective enthalpy of activation (Fig. 8h).

## Discussion

In photosynthetic water splitting, the protein determines structure and the spatial arrangement of the catalytic metal cluster with its inorganic $Mn_4CaO_{5/6}$ core and $Y_Z$, a nearby redox-active tyrosine sidechain. Aside from the basic structural role of the protein, the adjacent and more remote networks of H-bonded protein-water clusters facilitate efficient water oxidation by "orchestrating" the coupling between electron transfer steps and proton movements and likely also the insertion of the substrate-water molecules. Our study addresses three types of orchestrating protein roles, illustrated by analyses of the PSII variants with residue exchanges that affect the rate constants (reciprocal time constants) of the rate-limiting step of the oxygen-evolving $S_3 \rightarrow S_4 \rightarrow S_0$ transition.

### Coupling of electron and proton transfer

In the rate-limiting $S_3 \rightarrow S_4$ transition, the electron transfer likely is coupled to Grotthus-type PT from the substrate-hydroxide bound at Mn1 to the D61 carboxylate sidechain[10] (Fig. 1). Accordingly, a clear effect of the aspartate-alanine exchange in the D61A variant is expected and has previously been detected by time-resolved $O_2$-polarography[45], for the D61N variant also by time-resolved UV-vis

experiments[29]. We can now confirm the drastic slowdown of the $O_2$-evolution step by infrared spectroscopy. In light of the recent proposal of close coupling between $S_4$-formation and PT to D61 in the rate-determining step of the oxygen-evolution transition (ref. 10, see also refs. 43,44), it is surprising that $O_2$-evolution is slowed down by about two orders of magnitude but is not fully inhibited. Seemingly, the PT via the D61 sidechain is not obligatory but can be replaced by one or more alternative routes of coupling the $Y_Z$-reducing ET step to proton relocation; these are by about two orders of magnitude slower, but still allow for oxygen evolution. However, the oxygen evolution step proceeds at severely reduced transition efficiency (high miss factor, low amplitude of corresponding IR signals), explainable by competing recombination reactions during the strongly extended $Y_Z^{ox}$ lifetime. This illustrates the role of the protein in acceleration of the $O_2$-formation step to a level that facilitates an average quantum efficiency of the reactions in photosynthetic water oxidation well above 80% (miss factors well below 20%). Aside from providing relevant support to previous findings[25,30,45], the time-resolved IR data of the D61A variant is of importance in the present study because it facilitates comparison with the N298A variant.

## Orchestrating substrate-water binding

In the N298A variant, we encountered a paradox of severely reduced $O_2$-evolution under continuous illumination of saturating intensity but slightly accelerated $O_2$-release for excitation with short flashes of visible light. The time-resolved infrared spectroscopy supports resolution of this paradox by a plausible scenario.

The rate-determining proton-coupled ET step in the $S_3{\rightarrow}S_4$ formation as well as the subsequent processes of fast peroxide formation and $O_2$-release largely proceed in the N298A variant as they do in the wild-type, aside from the shift between entropy and enthalpy of activation discussed further below. These processes create a vacancy in the $Mn_4Ca$-oxo cluster that needs to be refilled to reach the $S_0$-state, by binding of a hydroxide to Ca and two five-coordinate $Mn^{III}$ ions (Ca, Mn3 and Mn4 in Fig. 6b)[16,46,65,66]. This is associated with the release of a proton[11,67,68] (Fig. 1a) stemming from coupled $H_2O$ deprotonation and hydroxide insertion into the vacant metal-cluster site. Despite its complexity, the $Mn_4Ca$-cluster reconstruction by water deprotonation and hydroxide insertion is faster than the rate limiting $S_3{\rightarrow}S_4$ transition in the wild-type PSII but not so in the N298A variant, where it is slowed down to about 75 ms (time-constant value). Consequently, only in the IR transients of the N298A variant the metal cluster reconstruction by hydroxide insertion is reflected as a separate kinetic phase. The time constant of 75 ms for completion of the $S_3{\rightarrow}S_0$ transition is in line with the strongly diminished continuous-illumination $O_2$-evolution in the N298A variant.

The N298 residue might affect the reactions in the oxygen-evolving $S_3{\rightarrow}S_4{\rightarrow}S_0$ transitions in different ways. First, its sidechain H-bonds the protonated $N_\delta$-H of the pivotal histidine-190 (H190). The complete loss of this H-bond likely would affect the proton affinity ($pK_a$) of the H190-$N_\epsilon$ severely, concomitantly the redox potential for $Y_Z$ oxidation, and consequently the rate constant of the rate-determining proton-coupled ET in the $S_3{\rightarrow}S_4$ transition. Experimentally, this is not observed, which our MD simulations can explain by the formation of an alternative H-bond to the peptide carbonyl of the neighboring residue (L297), which is present throughout a small fraction of the MD simulation time.

Second, the N298 sidechain lines the cavity of the so-called water wheel, a cluster of about 5 water molecules, which based on crystallographic data have been suggested to provide the $H_2O$ molecule for insertion into the $Mn_4Ca$-oxo cluster, in the $S_2{\rightarrow}S_3$ transition and likely also in the $S_4{\rightarrow}S_0$ transition[17,19,46]. Our findings suggest a drastic slow-down of the water insertion step in the $S_4{\rightarrow}S_0$ transition of the N298A variant, thereby providing experimental support for a crucial role of the water-wheel cluster.

Since extensive water movements proceed easily in the picosecond time domain (see Supplementary Movie 1), time constants in the millisecond domain do not relate to the water mobility itself, but rather to the Gibbs free energies of the initial state and the relevant transition state. Upon replacement of asparagine with a short alanine sidechain in the N298A variant, the water wheel gains space and expands towards the A298 sidechain (Fig. 7a). We hypothesize that, thereby, the free energy of the initial state of the water insertion step is lowered and consequently the activation energy increased, resulting in a major slow-down. However, also more complex explanations may apply.

The here-reported mutagenetic deceleration of later steps in the oxygen-evolving transition paves the road for more detailed spectroscopic and structural studies of these later steps, which previously were hidden behind the rate-determining $Y_Z^{ox}S_3{\rightarrow}Y_ZS_4$ transition. Further experimental studies on these kinetically hidden transitions will be highly valuable, or even indispensable, on the way towards a complete understanding of all aspects of photosynthetic water oxidation. Mutagenic deceleration may be an especially promising strategy but also in wild-type PSII, later steps might be resolvable[46].

## Balancing entropy and enthalpy of activation

The absolute rate constant value of the rate-determining step on photosynthetic oxygen evolution cannot be understood without considering a pronounced "entropic slow-down" likely assignable to multiple conformations of the H-bond protein-water network surrounding $Y_Z$ and the $Mn_4Ca$-oxo cluster of PSII[10,55]. We now detect a significantly increased entropic and reduced enthalpic contribution to the free energy of activation in the N298A variant that is caused by the exchange of a single residue only. The here-reported MD simulations with a force field developed for cyanobacterial PSII in the $Y_Z^{ox}S_3$ state provide evidence for an increased flexibility in key protein sidechains in the N298A variant and reduced probability to reach the transition state conformation, in line with an increased entropic contribution to the free-energy of activation of the $Y_Z^{ox}S_3{\rightarrow}Y_ZS_4$ transition. The increased flexibility suggests less stringent energetic constraints regarding nuclear movements, including the ones required to reach the transition state, which may explain the reduced enthalpy of activation.

## Relevance for inorganic (electro)catalysts

On first glance, it may appear far-fetched to compare the processes at the protein-bound $Mn_4Ca$ complex of PSII to inorganic catalyst materials. However, also the active phase of electrocatalyst materials based on first row transition metals typically comprises redox-active transition metals interconnected by bridging oxides and hydroxides[69–71], likely with extensive H-bonding of bridging and terminal $H_xO$ to nearby water molecules[72]; for Mn oxide catalysts, also a beneficial role of Ca ions has been demonstrated[73,74]. These materials are often discussed and addressed in computational studies by assuming simultaneous oxidation and deprotonation with focus on adsorption energies in various oxidation/protonation states, without considering the reaction kinetics (activation energies) of proton-coupled electron transfer, $O_2$-release, and substrate-water insertion (adsorption) in more detail. Previously, we showed that the slowest step in the reaction cascade of PSII water oxidation likely is the oxyl-radical formation by coupling the oxidation of a metal-bound hydroxide with its deprotonation in a Grotthus-type PT with an interesting relay functionality of a second metal ion[10]. In spite of its evolutionary optimization, this step is still associated with a sizeable free-energy of activation of about 580 meV (time constant around 2 ms). Now we provide evidence that in a genetic PSII variant, also the deprotonation-coupled binding (adsorption) of a substrate hydroxide may be associated with a sizeable free-energy of activation (about 670 meV, time constant around 70 ms). The magnitude of these two activation energies points to the need to consider increasingly more kinetic limitations (entropy and enthalpy

of activation) in electrocatalyst materials, which also in these materials could relate to multiple conformations and fluctuations of H-bond networks in vicinity of the catalytic site.

The rate-limiting processes can be addressed especially well for the Mn$_4$Ca-oxo cluster of PSII and its surrounding H-bonded water clusters because (i) experimentally, the transition between four semi-stable intermediates can be traced individually in time-resolved experiments and (ii) the well-defined active-site structure and adjacent water clusters favor detailed computational studies. Therefore, we propose to see the biological water oxidation as a model system that can inspire and direct the investigation of possibly analogous rate-limiting processes in electrocatalyst materials, and thereby the knowledge-guided development of new materials with diminished kinetic limitations.

## Methods

### Construction and propagation of mutants

The D1-D61A mutation has been described previously[40]. The D1-N298A mutation was created by changing the wild-type AAC codon of N298 to GCC in the *psbA*-2 gene of *Synechocystis* sp. PCC 6803[75]. The mutation-bearing plasmid was transformed into a *Synechocystis* strain that lacks all three *psbA* genes and has a hexa-histidine tag (His-tag) at the C-terminus of CP47[76]. Cultures were maintained and propagated as described previously[30]. To verify the integrity of the mutant cultures that were harvested for the purification of thylakoid membranes and PSII core complexes, the DNA sequence of the relevant portion of the *psbA*-2 gene was sequenced after PCR amplification of genomic DNA purified from an aliquot of each harvested culture. Purification of genomic DNA and PCR amplification of a 1380-bp fragment containing the entire *psbA*-2 gene was performed as described previously[75]. No trace of the wild-type codon was detected in any of the mutant cultures.

### Purification of thylakoid membranes

Harvested cells were suspended in 1.2 M betaine, 10% (v/v) glycerol, 50 mM MES·NaOH (pH 6.0), 5 mM CaCl$_2$, 5 mM MgCl$_2$, 1 mM benzamidine, 1 mM ε-amino-n-caproic acid, 1 mM phenymethylsulfonyl fluoride, and 0.05 mg/mL DNAse I, then broken in a glass bead homogenizer[30,77]. After removal of debris and unbroken cells by low-speed centrifugation, the resulting thylakoid membranes were concentrated by ultracentrifugation (20 min at 186,000 × *g* in a Beckman Ti45 rotor) and suspended in TM buffer [1.2 M betaine, 10% (v/v) glycerol, 50 mM MES·NaOH (pH 6.0), 20 mM CaCl$_2$, 5 mM MgCl$_2$] to a concentration of 1.0–1.5 mg of Chl/mL. The resulting membranes were either flash-frozen in liquid nitrogen or used immediately for the purification of PSII core complexes.

### Purification of PSII core complexes

Freshly prepared thylakoid membranes were extracted with *n*-dodecyl β-D-maltoside (at concentrations of 1 mg of Chl/mL and 1% (w/v) detergent) with gentle stirring in darkness for 10 min[30,77]. After unextracted material was removed by centrifugation (12 min at 35,300 × *g* in a Beckman JA-17 rotor), the supernatant was loaded onto a 40 mL, 5 cm diameter column of Ni-NTA superflow affinity resin (Qiagen, Inc., Valencia, CA) that had been equilibrated with PSII buffer [1.2 M betaine, 10% (v/v) glycerol, 50 mM MES·NaOH (pH 6.0), 20 mM CaCl$_2$, 5 mM MgCl$_2$, 0.03% (w/v) *n*-dodecyl β-D-maltoside]. After the column was washed with four bed volumes of PSII buffer at 5 mL/min, it was eluted with four bed volumes of PSII buffer containing 50 mM histidine at 5 mL/min. The eluted PSII core complexes were brought to 1 mM EDTA and concentrated by ultrafiltration. The purified PSII core complexes [in 1.2 M betaine, 10% (v/v) glycerol, 50 mM MES·NaOH (pH 6.0), 20 mM CaCl$_2$, 5 mM MgCl$_2$, 50 mM histidine, 1 mM EDTA, and 0.03% (w/v) *n*-dodecyl β-D-maltoside] were aliquoted, frozen in liquid N$_2$, and stored at 80 K.

### Mass spectrometry

PSII core complexes were extracted of lipids and pigments with methanol and ethyl acetate, then denatured with 5.2% lithium lauryl sulfate, 172 mM Tris-HCl (pH 8.0), and 40 mM dithiothreitol[78]. Denatured PSII subunits were separated by SDS-PAGE in gels containing 20% acrylamide and 6 M urea[26,78]. Excised D1 subunits were further denatured with urea (8 M), reduced with DTT (20 mM), alkylated with indoleacetic acid (50 mM), and buffer-exchanged into ammonium bicarbonate (50 mM). Subunits containing D1-D61A were digested with chymotrypsin for 16 h at 37 °C. Subunits containing D1-N298A were digested with trypsin for 16 h at 37 °C followed by partial digestion with chymotrypsin (2 h at 37 °C). Digested solutions were desalted with C18 Zip Tips (Waters Corporation), dried, and resuspended in 20 μL 0.1% formic acid. The resulting peptides were separated by nano-HPLC chromatography (EASY-nLC 1200, Thermo Scientific). Peptide samples (5 μL) were loaded onto a trap column (Thermo Scientific Acclaim PepMap C18, 75 μm × 2 cm) with a flow rate of 10 μl/min for 3 min, then separated on an analytical column (Acclaim PepMap C18, 75 μm × 25 cm) with a linear gradient of 3–37% buffer B (80% acetonitrile, 0.1% formic acid) in buffer A (0.1% formic acid) at a flow rate of 300 nL/min over 180 min. The column temperature was maintained at 45 °C. The nano-HPLC was coupled online with an electrospray tandem mass spectrometer (Orbitrap Fusion Tribrid Mass Spectrometer that was equipped with an EASY-Spray ion source). The electrospray voltage was 2.2 kV. The mass spectrometer was operated in data-dependent mode to switch automatically between MS and MS/MS acquisitions. MS spectra (m/z 375–1500) were acquired with a mass resolution of 60 K. High energy collisional dissociation (HCD) fragmentation was employed for recording MS/MS spectra. Raw data were processed and analyzed with MaxQuant (Version 2.1.4.0) with an in-house database containing the D1 protein and its mutants. Data from two preparations each of D1-D61A and D1-N298A core complexes were analyzed, each in comparison with wild-type PSII.

The mass spectrometry proteomics data have been deposited to the ProteomeXchange Consortium via the PRIDE[79] partner repository with the dataset identifier PXD068038.

### Steady-state O$_2$ evolution measurements

Initial light-saturated rates of O$_2$ evolution were measured at 25.0 °C in a solution of 1 M sucrose, 50 mM MES·NaOH (pH 6.5), 25 mM CaCl$_2$, and 10 mM NaCl that was held in a water-jacketed cell that was outfitted with a Clark-type oxygen electrode. For measurements, concentrated PSII core complexes (5 μg Chl) were diluted into 1.6 mL of assay buffer containing 1 mM potassium ferricyanide and 0.4 mM 2,6-dichloro-p-benzoquinone (the latter was purified by sublimation and diluted into the assay buffer immediately before the PSII core complexes were added). Red-filtered illumination was provided by two Dolan-Jenner (Woburn, MA) Model 180 fiber optics illuminators equipped with light guides. Error ranges were obtained as a standard deviation of 4–8 experiments.

### Time-resolved O$_2$ polarography measurements

An aliquot of thylakoid membranes (equivalent to 10 μg of chlorophyll) was suspended in buffer (150 mM NaCl, 25 mM MES, 1 M betaine, 5 mM MgCl$_2$ and 5 mM CaCl$_2$, adjusted to pH 6.2 with NaOH) and placed onto a bare platinum and silver-ring electrode[29], which was then centrifuged in a swing-out rotor at 10,000 × *g* for 10 min (at 4 °C). The polarization voltage of −0.95 V required for O$_2$-reduction at the platinum electrode was provided by a custom-built potentiostat and was switched on 15 s prior to the first excitation flash. For each of 80 flashes spaced by 900 ms, the current signal was recorded for a total of 500 ms (20 ms before and 480 ms after the flash). Slow drift contributions to the current signals were suppressed by a first-order high-pass filter (time constant of 100 ms); its influence was considered in simulation (fit) of the transients. Sample excitation was achieved by saturating flashes

(613 nm, 40 μs flash duration) from a red high-performance light-emitting diode (LED) operated at a maximal current density of around 150 A. The sample temperature was monitored by a miniature sensor within the sample buffer and controlled by Peltier elements.

A numerical diffusion model, inter alia, taking into account $O_2$ diffusion through the sample as well as the rate of $O_2$ consumption at the electrode, was used to accurately determine the rate constant of $O_2$ formation, as detailed elsewhere[29].

$O_2$ polarography experiments were performed at 8 distinct temperatures between 0 and 35 °C for wild-type PSII and for the N298A variant. At each temperature, at least three independent data sets were obtained and averaged before determining the time constants with the diffusion model; the given error is the one estimated by the fitting algorithm. The time constants were subjected to Arrhenius analysis (Fig. 2c), allowing for the determination of activation energy ($E_a$) and pre-exponential factor ($A$) by linear regression. The enthalpy and entropy of activation ($\Delta H^{\ddagger}$ and $\Delta S^{\ddagger}$) were calculated from $E_a$ and $A$ by applying Eyring's (and Polanyi's) transition state theory[52–54], using the equations provided as Supplementray Equations. The error estimations for $E_a$ and $A$ shown in Table 2 were obtained from the least-squares optimization process; the provided range of uncertainty for $E_a$ corresponds to the 1σ confidence interval of the slope value of the linear regression. As $\Delta H^{\ddagger}$ is calculated by subtracting a constant number from $E_a$, the error of $\Delta H^{\ddagger}$ is equal to the error of $E_a$. The 1σ confidence interval of $\Delta G^{\ddagger}$ was determined from the error range of the time constant $\tau_{O2}$ at 20 °C. The error of $T\Delta S^{\ddagger}$ is the sum of the errors of $\Delta G^{\ddagger}$ and $\Delta H^{\ddagger}$.

## Static FTIR difference spectroscopy

PSII core complexes were exchanged into a buffer containing 40 mM sucrose, 10 mM MES-NaOH (pH 6.0), 5 mM $CaCl_2$, 5 mM NaCl, and 0.06% (w/v) n-dodecyl β-D-maltoside, concentrated to approx. 3 mg of Chl/mL, mixed with 1/10 volume of fresh 100 mM potassium ferricyanide, spread into the center 13 mm of a 25 mm $BaF_2$ window, then dried lightly with $N_2$ gas[30,80]. The lightly dried samples were rehydrated to 95% relative humidity by placing six 1 μL drops of a solution of 40% (v/v) glycerol around the periphery of the sample[81]. A second window was placed over the first with a thin O-ring spacer in between. Sealed samples were equilibrated in the FTIR sample compartment at 0 °C in darkness for 1.5 h, illuminated with 6 pre-flashes, then dark-adapted for 30 min[30,80]. For each sample, the absorbance at 1657 cm$^{-1}$ (amide I band) was 0.6–1.1. Spectra were recorded with a Bruker Vertex 70 spectrometer (Bruker Optics, Billerica, MA) that held a pre-amplified, midrange D317 photovoltaic MCT detector (Kolmar Technologies, Inc., Newburyport, MA). Actinic flashes (~20 mJ/flash, ~7 ns fwhm) were provided by a frequency-doubled Q-switched Nd:YAG laser (BRIO, Quantel USA, Bozeman, MT). After dark adaptation, samples were given six flashes at 13 s intervals. Two transmission spectra were recorded before the first flash and one transmission spectrum was recorded starting 0.33 s after the first and subsequent flashes (each transmission spectrum consisted of 100 scans). The 0.33 s delay was included to allow the oxidation of $Q_A^-$ by the ferricyanide. Difference spectra of the successive S-state transitions (e.g., $S_{n+1}$-minus-$S_n$ difference spectra), were obtained by dividing the transmission spectrum obtained after the nth flash by the transmission spectrum obtained before the nth flash, then converting the ratio to units of absorption. The background noise level and the stability of the baseline were obtained by dividing the second pre-flash transmission spectrum by the first and converting the ratio to units of absorption (these spectra are labeled dark-dark in each figure—these are control difference spectra obtained in the presence of the sample but without a flash being given). The sample was then dark-adapted for 30 min and the cycle was repeated. For each sample, the illumination cycle was repeated 19 times. The spectra of 12–26 samples were averaged.

## Time-resolved single-frequency IR difference spectroscopy

PSII core complexes were left in the storage buffer but were concentrated to approximately 3 mg of Chl/ml and mixed with a 1/10 volume of fresh 100 mM potassium ferricyanide. About 10 μl of the sample solution was pipetted onto each of five 25 mm $CaF_2$ plates. The $CaF_2$ plates were prepared beforehand with a thin ring of vacuum grease and a 15 μm PTFE spacer. A second $CaF_2$ plate was placed on top of the first, sealing the sample in between. The samples were mounted onto an x-y-movable sample holder in a sample compartment, which was continuously flushed with dry air and kept at 10 °C. Per sample plate 39 individual spots were measured, resulting in a total of 195 sample spots. Prior to the first measurement the sample was illuminated with 2 pre-flashes and subsequently given an hour for dark adaptation. After being subjected to a flash sequence (6 s without excitation, followed by 10 sequential saturating flashes with 1 s flash spacing), each spot was given an hour to dark adapt before being measured again. The 10th excitation flash was about three times stronger than the first nine flashes, which allows for the estimation and correction of a flash-induced heat artifact as described previously[10,57]. The excitation flashes were provided by a frequency-doubled Q-switched Nd:YAG laser (Minilite II, Amplitude Laser, Inc., Milpitas, CA). The IR source was a QCL covering the wavenumber range of 1650 – 1300 cm$^{-1}$ (MIRcat, Daylight Solutions, Inc., San Diego, CA). The transmitted IR signal was measured with a 10 MHz pre-amplified MCT detector (Vigo Systems, Poland) and a second, identical detector was used as reference to improve the overall signal-to-noise (S/N). The intensity of the excitation flashes was measured by a power meter and a photodiode was used to determine the timing of the flashes. Data processing and analysis was performed in Python 3.7. An artifact signal of unknown origin was found in the millisecond region of the transients; as the same artifact was also recorded by the reference detector, we were able to correct for this artifact.

For analysis of the $S_3{\rightarrow}S_0$ transition of wild-type PSII and the N298A variant, the flash-induced IR transients were deconvolved into transients corresponding to the individual S-state transitions, in a similar manner as e.g., described in refs. 10,82. For this an initial starting population of 90% $S_1$ and 10% $S_0$ were assumed. The miss factor of each dataset was determined by simulating the flash-dependency of the IR difference signals and was estimated to be 10–15% for wild-type PSII and 12–18% for N298A. A deconvolution of the D61A data was not (satisfactorily) possible due to the strongly retarded behavior of this variant.

To analyze the kinetics, the transients were fit to a sum of exponentials:

$$y_{sim}(t) = \sum_i A_i{}^*(1 - e^{-t/\tau_i}) + y_0 \qquad (1)$$

For each PSII variant, the time constants ($\tau_i$) were determined globally over all four wavenumbers, while the amplitudes ($A_i$) and the offset ($y_0$) were left to vary freely. The parameters were optimized by a least-squares algorithm. The error of the parameters was estimated by performing fits of multiple subsets of the four transients and calculating the standard deviation. For more details see Section 4 in the Supporting Information.

## Classical molecular dynamics simulations

As initial structure for our simulations we have considered the cryo-EM structure of *Synechocystis* photosystem II resolved at 1.93 Å resolution (Protein Data Bank ID 7N8O)[64].

MD simulations were performed using the GROMACS package[83]. The dimeric complex was solvated in a water box with dimensions $31.3 \times 20.9 \times 15.3$ nm and embedded into a POPC lipid bilayer, following the general scheme described in refs. 84,85. All simulated systems were composed of approximately 1,000,000 atoms.

In total, ten classical MD simulations were performed to investigate the wild-type protein and the N298A variant. Five unrestrained 500 ns simulations were run for each system in the NPT ensemble with different starting velocities (referred to as WT-1, WT-2, WT-3, WT-4, WT-5, N298A-1, N298A-2, N298A-3, N298A-4, and N298A-5). Additional details on the methodology, including system setup, parameterization, and simulation protocols, are available in the Supplementary Information (Molecular Dynamics section).

### Reporting summary

Further information on research design is available in the Nature Portfolio Reporting Summary linked to this article.

## Data availability

The time-resolved IR data obtained for 10 consecutive excitation flashes (applied to wild-type PSII, N298A and D61A) are available at Zenodo[86] [https://doi.org/10.5281/zenodo.15789664]. The mass spectrometry data are available via ProteomeXchange with identifier PXD068038. Source data are provided with this paper.

## Code availability

Python code (Python 3.7.16) was used to correct the time-resolved IR data for S-state starting populations and cycle inefficiency (miss factor) and to fit the data globally to a sum of exponentials. This code is analogous to code deposited previously (https://doi.org/10.5281/zenodo.7682034). Standard GROMACS packages were used for the MD simulations, which are generally available—see also ref. 83.

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

## Acknowledgements

S.M.M., R.A., P.S.S., and H.D. gratefully acknowledge financial support by the Deutsche Forschungsgemeinschaft (DFG, German Research Foundation) provided to the collaborative research center on Protonation Dynamics in Protein Function (SFB 1078, project A4/ Dau) and under Germany's Excellence Strategy—EXC 2008— 390540038-UniSysCat. G.P., C.D.S., D.N. and L.G. acknowledge the CINECA award under the ISCRA initiative (project IsB29_PSIICMD, D.N.), for the availability of high-performance computing resources and support, and would like to especially thank Paola Alberigo for her valuable assistance in facilitating and extending the computational resources needed for this work. This work was supported by the Department of Energy, Office of Basic Energy Sciences, and Division of Chemical Sciences Grant DE-SC0005291 to R.J.D. We thank Dr. Quanqing Zhang, IIGB Proteomics Manager at UC Riverside, for the protein mass spectrometry analyses.

## Author contributions

S.M.M. collected, analyzed, and conceptualized the time-resolved IR data; preliminary data was collected by P.S.S. G.P., C.D.S. and D.N. performed the computational research. R.A. collected and analyzed the time-resolved $O_2$ polarography data. R.J.D. constructed the site-directed PSII mutants, purified PSII complexes, collected and analyzed the FTIR data. H.D., R.J.D. and L.G. designed and conceptualized the experimental and computational research. H.D., S.M.M., R.J.D., G.P., D.N. and L.G. wrote the paper.

## Funding

## Competing interests

The authors declare no competing interests.
