## [Transparent Peer Review file · Nature Communications]

Three rate-determining protein roles in photosynthetic O₂-evolution addressed by time-resolved experiments on genetically modified photosystems

Corresponding Author: Professor Holger Dau

Version 0:

Reviewer comments:

Reviewer #1

(Remarks to the Author)

The manuscript studies kinetic aspects of the last step and resetting transition (S₃-S₄-S₀) in the catalytic cycle of nature's water oxidizing complex OEC. The focus is on two PSII mutants, D61A and N298A, mostly the second. Combination of time-resolved O₂-polarography, FTIR difference spectroscopy, time-resolved IR are used to determine differences between mutants and wild-type (WT) PSII. Greife et al. (Nature 2023) had studied the S₃-S₀ sequence in WT PSII and determined time constant of key steps in S₃-S₄ because S₄-S₀ is too fast after O₂ formation to be kinetically resolvable. The finding here is that in N298A mutant the oxygen evolution step (1st process) is equally fast as in WT, but events that follow O₂ formation/evolution up to reconstitution of S₀ state (2nd process, S₄-S₀) are slower in the mutant. Significant deceleration of the second step makes it kinetically detectable in N298A, perhaps the first kinetic detection of processes after O₂ evolution. An interesting point is made about entropic/enthalpic contributions to the first step echoing past discussion in the WT. Compensation between the two terms results in same time constants between mutant and WT. Ideas are proposed about possible mechanistic interpretations. The manuscript incorporates a part with molecular dynamics on the mutant. This is a good specialized study, expertly conducted, that contributes useful new data on kinetic aspects of the mechanism.

On the other hand, insight beyond the kinetics is limited because structural interpretations are hypothetical. Molecular origins of the observations remain unknown because mutant structures and relevant mechanistic steps are unknown. The discussion is based on assumptions about structure, identity of specific steps (Figure 6) and role of components e.g. "water wheel". Although such ideas are inspired from the literature they have no direct support. The S₃→S₀ sequence in Figure 6 is plausible hypothesis and should be treated as such rather than as a new finding. It is necessary to limit the creative aspects of the manuscript in a possible scenario section and separate speculation from results in the text. Same for the conclusions where data and hypothetical interpretations overlap.

Specific points to consider:

1. D61A results are confirmatory. Helpful for comparisons, but perhaps redundant. If missing from the manuscript, the results and discussion about WT and N298A would be the same. Unclear how interpretations about D61A follow from the data, especially if time constants of 15 ms and 360 ms remain chemically undefined.
2. Time constants are measured on absorption changes of four wavenumbers: 1384 cm⁻¹, 1400 cm⁻¹, 1514 cm⁻¹ and 1544 cm⁻¹; it is not explained why were these chosen and which modes are responsible, not discussed in p. 9-10. In the previous work 1381 cm⁻¹, 1571 cm⁻¹, and 1707 cm⁻¹ were used, why?
3. It would be helpful in time-resolved IR (Figure 5a) to note the times that correspond to phase changes in the diagrams.
4. Line 383: "In the WT-PSII [the process in the oxygen evolving transition that occurs after O₂-release] is faster than the O₂-evolution step and thus not resolvable in time-resolved experiments". However, in Figure 5b (and Table S2) two time constants are given for the WT 4.7 ms and 60 ms. Clarify what you think is happening in WT in the 60 ms.
5. Is τ₃ (in S1) supposedly the time constant for deprotonation of S3YZox as in previous paper?
6. D61A: what are the times? Why Fig. 6 is labeled "severely slowed down" while the first time constant is 15 ms? Why is the second (after O₂ evolution) "invisible", while the second time constant is 360 ms?
7. The computational part confirms expectations but does not add insight. Conclusions of MD regarding perturbations of the "water wheel" and Hyd.-bonds around H190 (Figure 7b) are already expected from the known WT structure. Can MD simulate how mutants fold in the first place? Probably not, so models in Figure 7 are uncertain. Validation of computation

- can be made with predictions of data discussed in the paper, vibrational modes, kinetic parameters, entropic/enthalpic terms.
8. It is suggested that in 50% Yz oxidation is more difficult: is this apparent from experiment?
 9. I am not convinced that visual inspection of water positions (Figure 7) can lead one to correlation with entropic terms of a process that is not studied in this work and not observed in crystallographic studies.
 10. Lines 440-441: "The main effect of the N298A modification may be described as a less constrained 440 conformational space in the vicinity of this residue." Quantification of "more/less constrained" is required. In Table S4 the average number of water molecules is identical in WT PSII and N298A. Other aspects such as PCET would be foremost affected but kinetic data on the first step do not suggest this. Entropic and enthalpic contributions fitted and given in Table 2 refer to the oxygen evolution step so there is mismatch in points discussed. No data available on entropic contribution in the phase after O₂ formation.
 11. Lines 460-461: reasonable statement but does not follow from data because no experiments probe H190.
 12. Lines 518-520: the statement summarizes the weakness of the presentation: "our findings suggest a drastic slow-down of the water insertion step...": in fact they only suggest the presence of an unknown bottleneck. "... thereby providing experimental support for a crucial role of the water-wheel cluster": no such connection can be deduced, this is a very long stretch of interpretation.
 13. It is awkward that the opposites of hypothetical explanations for the effect of mutations are presented as "functions" of the WT, "roles" in the title.

Reviewer #2

(Remarks to the Author)

Reviewer #3

(Remarks to the Author)

This manuscript by Mausle and coworkers describes a diverse and complementary series of experiments that aim to understand how protein environment affects water oxidation and O₂ evolution in Photosystem II (PSII). Two point mutations are studied using the model cyanobacterium *Synechocystis* 6803. The rate of O₂ evolution following a single turnover flash is measured using polarography at multiple temperatures to facilitate Arrhenius analysis. Static and time-resolved FTIR are used to probe the protein environment and accompanying water molecules. Finally, molecular dynamics (MD) simulations are performed to support the experimental results. The manuscript is very well written, and the figures are well designed. Overall, the work is thorough, expertly conducted (with one concern described below), and provides profound insights into PSII O₂ evolution mechanisms.

My only significant concern with this manuscript is the discrepancy between the use of mesophilic *Synechocystis* in experimental studies and thermophilic *Thermosynechococcus* in MD studies. A high resolution cryo-EM structure of *Synechocystis* PSII has been available since 2021 (Gisriel, et al. 2021, PNAS). While I am typically hesitant to demand additional experiments, the manuscript would be much stronger if the MD simulations were performed on a structure based on that of *Synechocystis*. H-bond networks in PSII are not identical between mesophilic and thermophilic cyanobacteria. Therefore, the presented MD data are not convincing.

A more minor point concerns the relationship between Figures 3 and 4. Please edit the text to connect these datasets more. As a non-FTIR expert, it is confusing why different wavenumbers are selected for discussion in the static data and for analysis in the time-resolved data.

Three simple points:

Check the Chu, et al. citation in the caption of Table 1. Add that the WT O₂ rate in that work was 680 $\mu\text{mol}/\text{mg}/\text{h}$.

Please tone down the usage of words such as astonishing, stunning, remarkable, etc. Using these terms more sparingly will increase their impact.

Figure S14 nicely shows distance fluctuations in the N298A mutant that are absent in WT. If the data are reproduced using the *Synechocystis* structure, consider moving this figure (or a smaller version of the figure) to the main text.

Reviewer #4

(Remarks to the Author)

Review report of manuscript "Three rate-determining protein roles in photosynthetic O₂-evolution addressed by time-resolved experiments on genetically modified photosystems".

Mäusle and coworkers studied the time-evolved OEC complex by time resolved IR spectroscopy. The OEC evolving during water splitting problem has been attracted for many years and the long-standing question could be always the uniqueness of the refined model and parameters. The X-ray time resolved spectroscopy and diffraction would be another nice approach for the study, except the damage problem. The authors proposed to employ the TRIR spectroscopy for this study and it has

been investigated by many groups in these years. However, the mutation of samples highlighted the novelty of this work. This manuscript contains important and interesting results, the novelty could make it publishable in Nature Comm. However, the current version needs to be refined/modified before the acceptance.

(1). The same question, the uniqueness of the mechanism proposed by authors. I would like to suggest the authors to do some test works, assuming different mechanism with different structural changes, to demonstrate the uniqueness of the model and parameters. The detailed descriptions can be listed in the revised SI and should add a short paragraph to describe in the discussion part of the manuscript.

(2). It could be useful if authors to plot the time-resolved IR spectra as a three-dimensional data, for instance, the x as the time, y axis as the lambda and the color bar, as the normal plotting of TA data. It would be useful to perform the global fitting to retrieve the lifetime components? More informative signals could obtain with decay associated spectrum and their decay timescales. The spectral profile could provide more valuable information, rather than the single-trace analysis.

(3). Do you think if this can be measured at lower temperature? The spectroscopic signals could be more informative, OK, maybe the flashing sample is hard to be achieved? Any idea to perform at low temperature? I knew the protein crystal and X-ray diffraction has been measured at lower temperature. For instance, they called it "chip" technology in Hamburg, which allows the researcher to diffract each crystal with a chip loading sample.

(4). In Fig4, I never expect to have such amount of changes in signal, and even with the error bars. Or maybe I am not in the field for long time. Did the group perform in an advanced spectrometer or something wrong? Could you please explain it in details?

(5). Except Fig. 1, Fig6 and Fig. 7, the rest of figures need to be refined. For instance, Fig. 3 looks quite mess. The authors need to consider to plot them in a better format, to meet the Nature Comm. Style.

(6). How about the X-ray diffraction? Is possible to perform these samples in the diffraction? Or electron diffraction. I would like to hear from authors about the studies with diffractive approaches? Or X-ray absorption spectroscopy?

Version 1:

Reviewer comments:

Reviewer #1

(Remarks to the Author)

The revised version addresses comprehensively all points raised initially, and substantially clarifies all issues that might have created confusion in the first version. The presentation is sharpened, the results and arguments are easier to follow. New extensive MD simulations offer additional insights and support, while the more hypothetical interpretations are better demarcated in the discussion. A couple of final comments regarding N298: is my understanding correct, that there is no expectation that the N298A mutation would affect the Yz redox potential? And could the authors comment on the hypothesis of involvement of N298 in proton translocation (Chrysina et al. J. Phys. Chem. B 2019, 3068) or whether the EPR observations can be reinterpreted in light of the more likely participation of this residue in water insertion?

Reviewer #2

(Remarks to the Author)

Reviewer #3

(Remarks to the Author)

My concerns have been fully addressed. The authors performed extensive new calculations which improve the manuscript.

Reviewer #4

(Remarks to the Author)

I have read the reply letter and the revised manuscript from authors and I think almost all my questions have been well answered. Especially, authors have treated the figures and replotted/reformatted the new figures for manuscript. I think the current version of manuscript and the supporting information could match the quality of publication.

Response to the

REVIEWER COMMENTS

[Reviewer comments using black letters, our response using blue letter.]

We thank the reviewers for their constructive comments, which have significantly contributed to improving our manuscript. We hope they will find the thoroughly revised version both informative and satisfying. This revision includes substantial new data and analyses, as outlined below.

In response to the reviewers' feedback, we have made extensive clarifications, incorporated additional computational data, and carried out further analyses of the experimental results. Notably, we have now performed MD simulations for the cyanobacterial PSII from *Synechocystis* PCC 6803—the actual species investigated experimentally—extending the total trajectory duration to 5 μ s. The computational demands of these simulations, conducted on dedicated supercomputing facilities, along with the subsequent analyses, account for the time required to complete this revision. The new simulation results are presented in Figures 7 and 8, with additional computational data in several supplementary figures. Supplementary Fig. 20 provides a reassuring comparison between our MD results for two cyanobacterial PSII species and corresponding structural data from crystallography and cryo-EM. We also present additional time-resolved IR data and new spectroscopic data analyses in the Supplementary Information and clarify the reaction sequence by simulating the population distributions of possible intermediates using experimentally determined rate constants (see Figure 5).

Further details are provided in our point-by-point response to the reviewers' comments. Additionally, we have included more detailed information on error estimation and other technical aspects, along with various minor clarifications. These and all other substantive changes are highlighted in red in the revised manuscript; typographical corrections are not marked.

Reviewer #1 (Remarks to the Author):

The manuscript studies kinetic aspects of the last step and resetting transition (S3-S4-S0) in the catalytic cycle of nature's water oxidizing complex OEC. The focus is on two PSII mutants, D61A and N298A, mostly the second. Combination of time-resolved O₂-polarography, FTIR difference spectroscopy, time-resolved IR are used to determine differences between mutants and wild-type (WT) PSII. Greife et al. (Nature 2023) had studied the S3-S0 sequence in WT PSII and determined time constant of key steps in S3-S4 because S4-S0 is too fast after O₂ formation to be kinetically resolvable. The finding here is that in N298A mutant the oxygen evolution step (1st process) is equally fast as in WT, but events that follow O₂ formation/evolution up to reconstitution of S0 state (2nd process, S4-S0) are slower in the mutant. Significant deceleration of the second step makes it kinetically detectable in N298A, perhaps the first kinetic detection of processes after O₂ evolution. An interesting point is made about entropic/enthalpic contributions to the first step echoing past discussion in the WT. Compensation between the two terms results in same time constants between mutant and WT. Ideas are proposed about possible mechanistic interpretations. The manuscript incorporates a part with molecular

dynamics on the mutant. This is a good specialized study, expertly conducted, that contributes useful new data on kinetic aspects of the mechanism.

We appreciate the precise and concise summary of our work and its positive assessment.

On the other hand, insight beyond the kinetics is limited because structural interpretations are hypothetical. Molecular origins of the observations remain unknown because mutant structures and relevant mechanistic steps are unknown. The discussion is based on assumptions about structure, identity of specific steps (Figure 6) and role of components e.g. “water wheel”. Although such ideas are inspired from the literature they have no direct support. The $S_3 \rightarrow S_0$ sequence in Figure 6 is plausible hypothesis and should be treated as such rather than as a new finding. It is necessary to limit the creative aspects of the manuscript in a “possible scenario” section and separate speculation from results in the text. Same for the conclusions where data and hypothetical interpretations overlap.

We thank the reviewer for the insightful and constructive suggestions, which we have addressed as detailed in the following.

Separation of results and hypotheses

We have followed the important suggestion of the reviewer to separate more clearly the experimental results from hypothetical aspects by discussing the latter as a scenario. Therefore, we introduced a section with the headline:

“Population of intermediate states in the $S_3 \rightarrow S_4 \rightarrow S_0$ transition – a possible scenario”

After a summarizing statement, this section starts with an additional new figure to clarify the proposed reaction scheme and the time dependence of population of intermediates without providing atomistic details (Fig. 5). In a second step, the atomistic details are shown in Figure 6.

We would like to point out that the hypothetical character is lower than implied by the reviewer. We believe that the reaction scheme in Figure 5A is the only plausible one that can account for the results of previous and the present studies straightforwardly. Noteworthy, in these previous studies a variety of complementary time-resolved experimental methods were used, inter alia time-resolved X-ray absorption spectroscopy with its clear sensitivity to the oxidation of the Mn ions and structural changes of the metal complex.

The atomistic details shown in Figure 6 come from structure-based computational investigations that have been published previously and do represent, in our opinion, a valuable illustration to the readers, albeit being hypothetical in particular regarding panel b. This is now clarified by the following sentence in caption of Figure 6: “The arrangements of the non-hydrogen atoms schematically shown in panels a and c correspond to crystallographically resolved structures^{16,45}. The structures in panel b, the location of H-atoms, and all the particle movements indicated by arrows have been deduced by structure-based computational chemistry previously¹⁰ and thus are more hypothetical than the arrangement of non-hydrogen atoms shown in panels a and c.”

Structural changes induced by the N298A exchange

Now, in the revised manuscript, the possible structural consequences of the N298A exchange are investigated computationally for the PSII of the species actually used in our experimental study, namely *Synechocystis* PCC 6803 (in the originally submitted manuscript the MD simulations were done for *Thermosynechococcus vestitus*). We find plausible conformational changes in the vicinity of the exchanged residue and increased flexibility at slightly larger distances (of the E189 sidechain), but not any more major changes were detected over all the MD trajectories. This concurs with the largely unmodified steady-state FTIR difference spectra, which essentially excludes significant changes in structure and ligand environment of the Mn₄Ca-oxo complex. Regarding the average number and position of water molecules, reassuringly good agreement with crystallographic results is obtained for both *T. vestitus* PSII and the wild-type PSII of *Synechocystis* PCC 6803, see Supplementary Figure 20, confirming the predictive power of the MD simulations in this regard and providing confidence in the results obtained for the genetic variant (N298).

Specific points to consider:

1. D61A results are confirmatory. Helpful for comparisons, but perhaps redundant. If missing from the manuscript, the results and discussion about WT and N298A would be the same. Unclear how interpretations about D61A follow from the data, especially if time constants of 15 ms and 360 ms remain chemically undefined.

We see the point of the reviewer but still believe that presenting the D61A data in comparison to the WT and N298A data will be useful to the reader. We want to show that the functioning of the D61A mutant is strongly affected by the residue exchange, but the residue exchange effect differs starkly from that of the N298A variant in a way that can be plausibly rationalized as shown in the new (additional) Figure 5. In any event, we are thankful to the reviewer for pointing out that our presentation in the originally submitted manuscript was lacking the clarity to see the benefits of presenting also the D61A data. In the revised manuscript, we address the reviewer's concern by clarifications in the article, inter alia by means of Figure 5, and in the SI.

Noteworthy, the 360 ms rate constant and the corresponding infrared signals are not chemically unassigned. They correspond to the processes described in the WT by the 5 ms rate constant, but are drastically slowed down in the D61A variant, in line with the previously published time-resolved O₂-release data and UV/vis data and inter alia supported by identical +/- amplitude signs of this phase at all four wavenumbers when compared to the WT. The chemical assignment of the 15 ms phase, however, is indeed unclear. This phase is not specific to the S₃→S₀ transition, but is present at even larger amplitude also in the S₂→S₃ transition of D61A. This point is now stated in the text and supported by additional material and discussion in the SI (Supplementary Figs. 16-18 and corresponding Supplementary Discussion).

2. Time constants are measured on absorption changes of four wavenumbers: 1384 cm⁻¹, 1400 cm⁻¹, 1514 cm⁻¹ and 1544 cm⁻¹; it is not explained why were these chosen and which modes are responsible, not discussed in p. 9-10. In the previous work 1381 cm⁻¹, 1571 cm⁻¹, and 1707 cm⁻¹ were used, why?

The selection of wavenumbers and their putative assignment is now explained in the revised manuscript on pg. 10, 1st and 2nd paragraph under the headline “Time-resolved IR reveals biphasicity in the O₂-evolution transition of D1-N298A”.

3. It would be helpful in time-resolved IR (Figure 5a) to note the times that correspond to phase changes in the diagrams.

The time constant values are now indicated in Figure 4 by colored arrows.

4. Line 383: “In the WT-PSII [the process in the oxygen evolving transition that occurs after O₂-release] is faster than the O₂-evolution step and thus not resolvable in time-resolved experiments”. However, in Figure 5b (and Table S2) two time constants are given for the WT 4.7 ms and 60 ms. Clarify what you think is happening in WT in the 60 ms.

This time constant is not specific for the S₃→S₄→S₀ transition, likely assignable to an acceptor side contribution as now discussed in more detail (along with additional data) in the Supplementary Information (Supplementary Figs. 16-18 and associated Supplementary Discussion).

5. Is τ₃ (in SI) supposedly the time constant for deprotonation of S₃Y_Zox as in previous paper?

Correct, τ₃ has been assigned in several previous investigations to the deprotonation step induced by formation of S₃Y_Z^{ox}. We now have included the corresponding time constant and amplitude in Figure 4 and clarify its assignment to an essential deprotonation step preceding oxygen evolution in Figure 5. We thank the reviewer for pointing out the need for clarification.

6. D61A: what are the times? Why Fig. 6 is labeled “severely slowed down” while the first time constant is 15 ms? Why is the second (after O₂ evolution) “invisible”, while the second time constant is 360 ms?

Please see our response to the specific point No. 1 provided further above.

7. The computational part confirms expectations but does not add insight. Conclusions of MD regarding perturbations of the “water wheel” and Hyd.-bonds around H190 (Figure 7b) are already expected from the known WT structure. Can MD simulate how mutants fold in the first place? Probably not, so models in Figure 7 are uncertain.

The MD simulations cannot track the folding of the protein, but they can confirm the stability of a given protein structure. This point is strengthened in the revised manuscript by further MD simulations which are now done for PSII of *Synechocystis* PCC 6803, wild-type and N298A variant, the species used in the present investigations. These simulations indicate a significant effect on the extension of the “water wheel” as well as more flexibility of some crucial residues in the vicinity of the mutated residue, in particular the His190 (and its H-bonding partner) and the Glu189; please see Figures 7 and 8 of the revised manuscript and the related discussion in the article. The overall structure of PSII and its co-factors is shown to be stable throughout all the MD trajectories. This is in line with the largely unchanged FTIR difference spectra, which evidence the absence of modifications of the ligand sphere of the manganese-calcium cluster of PSII.

We also see a major heuristic value of these calculations by illustrating the surprising dynamic variability of the water clusters, with occupancies of the water-wheel region ranging from 2 to 7 water molecules. By similar calculations it may become possible in the future to understand better what the “perturbations” visible in structural data could mean. Inter alia for this reason,

structural data on the N298A would be highly welcome, in particular in terms of crystallographic snapshots that track advancement through the S-state cycle, but this clearly is beyond the scope of our present study.

In the revised manuscript, the MD simulations are coupled to a more detailed quantitative analysis and improved presentation in the form of two new figures in the main manuscript.

8. It is suggested that in 50% Yz oxidation is more difficult: is this apparent from experiment?

This seems to be a misunderstanding. It is not suggested that 50% Yz oxidation is more difficult. The kinetic scheme and population distributions in Figure 5 of the revised manuscript may clarify this point.

9. I am not convinced that visual inspection of water positions (Figure 7) can lead one to correlation with entropic terms of a `_process_` that is not studied in this work and not observed in crystallographic studies.

In the revised manuscript, *the visual inspection of the water molecules now has been complemented by a quantitative analysis shown in Figure 7. We note that this result of the MD simulation is not meant to address specifically the entropic contribution to activation energy. These are rather the results shown in Figure 8 of the revised manuscript. We hope that the clarity of presentation is improved by presenting the MD in two figures that address different aspects of the experimental observations, namely Figure 7, the water wheel extension and dynamics and Figure 8, conformational dynamics possible related to the experimentally observed entropy-enthalpy compensation.*

10. Lines 440-441: “The main effect of the N298A modification may be described as a less constrained 440 conformational space in the vicinity of this residue.” Quantification of “more/less constrained” is required. In Table S4 the average number of water molecules is identical in WT PSII and N298A.

We have addressed this point by additional quantitative analysis, see Figure 7 of the revised manuscript. (We note in passing that the numerical values for the average number of water molecules within the water wheel regions changed slightly for MD simulation now done for *Synechocystis* PCC 6803 over an even more extended time period, without affecting our conclusions. See also the new version of Supplementary Table 5, formerly Table S4.)

Other aspects such as PCET would be foremost affected but kinetic data on the first step do not suggest this. Entropic and enthalpic contributions fitted and given in Table 2 refer to the oxygen evolution step so there is mismatch in points discussed. No data available on entropic contribution in the phase after O₂ formation.

The discussed entropy-enthalpy compensation indeed relates to the activation energy of the rate-limiting step in O₂-formation. A similar entropy-enthalpy compensation is found when comparing plant PSII and cyanobacterial PSII, as discussed elsewhere (see Dau and Greife, 2023, *Inorganics*). Now we show that this shift between the entropic and enthalpic contribution can be induced by a single residue exchange, which came as a true surprise to us. This effect of a single residue exchange opened the option for comparative investigation by MD simulations, with interesting results that support a general softening of the relevant energy surface by this single-site exchange, see Figure 8, in line with the observed shift in the entropy-enthalpy as now qualitatively illustrated in the energy scheme of Figure 8h. *However, we do not suggest that the entropy-enthalpy compensation effect explains the slow-down of the water insertion step in the*

N298A mutant, for which we did not investigate its temperature dependence. *These are two separate phenomena, but both caused by the N298A mutation.* (Meanwhile we know that also other single-site mutations affect the entropy-enthalpy ratio, to different extents; further computational analysis of this interesting effect is planned.)

By discussing the two effects originating from the N298A exchange as two different rate-determining roles of the protein, we hope to clarify that the distinction between the two effects. Moreover, in the revised manuscript, the computational data now is presented in form of two separate figures, (i) Figure 7 addressing the possible role of the “water wheel” in the slow-down of the water insertion step and (ii) Figure 8 showing the softening of the energy landscape along the putative reaction coordinates in the N298A mutant.

11. Lines 460-461: reasonable statement but does not follow from data because no experiments probe H190.

We thank the referee and agree that, while the conformational variability of the H190 sidechain is evident from our calculations, we cannot firmly evaluate its impact on the pKa. Therefore, these lines have been modified by removing this statement from the revised manuscript.

12. Lines 518-520: the statement summarizes the weakness of the presentation: “our findings suggest a drastic slow-down of the water insertion step...”: in fact they only suggest the presence of an unknown bottleneck. “... thereby providing experimental support for a crucial role of the water-wheel cluster”: no such connection can be deduced, this is a very long stretch of interpretation.

We have rephrased or omitted the criticized statements and generally approached improved clarification of the hypothetical character. However, as explained in our response to the general comments of the reviewer, it is not the case that we merely provide evidence for an unknown bottleneck step. There is sufficient experimental evidence to justify our suggestion of a drastic slow-down of the water insertion step (or/and the coupled water deprotonation). In the revised manuscript, the evidence is summarized in the first paragraph of the scenario section on pg. 13/14, leading to the following summarizing statement: “Since the slow process in the N298A variant occurs after O2-release, we assign it to the water deprotonation and hydroxide insertion needed to refill the vacant O5 site, finally resulting in the Mn4CaO5 cluster in its S0 state. Which of these steps, water deprotonation and insertion, represents the rate-limiting bottleneck remains open as does their sequence; likely these processes proceed in a tightly coupled way.”

Our study does not provide clear-cut, but rather circumstantial support for a crucial role of the “water wheel”; accordingly, our corresponding statements are phrased such that the hypothetical character is not obscured. Nonetheless, it should not be overlooked that the new computational results and their analysis now summarized in Figure 7 support a possible water-wheel influence on the experimentally observed slow-down of the water insertion step in the N298A variant.

13. It is awkward that the opposites of hypothetical explanations for the effect of mutations are presented as “functions” of the WT, “roles” in the title.

We understand what the reviewer is aiming at but still do think that the expression of “rate-determining roles of the protein environment” is adequate when considering the emphasis on “rate-determining”. Indeed, none of the investigated protein variations fully inhibit oxygen evolution, but the rate constants are affected (severe slow-down is of the rate-determining step

plus a change in the enthalpic/entropic contributions to the free-energy of activation). This implies that the evolutionary optimized protein environment indeed accelerates the chemical reactions which otherwise may proceed at rates that are too slow for competing with charge recombination, as now explicated for the D61A variant on pg. 20:

“.. the oxygen evolution step proceeds at severely reduced transition efficiency (high miss factor, low amplitude of corresponding IR signals), explainable by competing recombination reactions during the strongly extended YZox lifetime. This illustrates the role of the protein in acceleration of the O₂-formation step to a level that facilitates an average quantum efficiency of the reactions in photosynthetic water oxidation well above 80% (miss factors well below 20%).”

The role of the protein environment in accelerating reactions which otherwise would proceed at low rates also gains meaning when considering that in analogous inorganic catalyst materials the turnover frequencies are typically lower by orders of magnitude than in PSII. So, our study also highlights the challenge to mimic or substitute the accelerating roles of the protein environment in inorganic catalyst materials.

Reviewer #2 (Remarks to the Author):

Thank you for your efforts and participation in this important initiative of Nature Communications (or Nature-Springer in general).

Reviewer #3 (Remarks to the Author):

This manuscript by Mausle and coworkers describes a diverse and complementary series of experiments that aim to understand how protein environment affects water oxidation and O₂ evolution in Photosystem II (PSII). Two point mutations are studied using the model cyanobacterium *Synechocystis* 6803. The rate of O₂ evolution following a single turnover flash is measured using polarography at multiple temperatures to facilitate Arrhenius analysis. Static and time-resolved FTIR are used to probe the protein environment and accompanying water molecules. Finally, molecular dynamics (MD) simulations are performed to support the experimental results. The manuscript is very well written, and the figures are well designed. Overall, the work is thorough, expertly conducted (with one concern described below), and provides profound insights into PSII O₂ evolution mechanisms.

My only significant concern with this manuscript is the discrepancy between the use of mesophilic *Synechocystis* in experimental studies and thermophilic *Thermosynechococcus* in MD studies. A high resolution cryo-EM structure of *Synechocystis* PSII has been available since 2021 (Gisriel, et al. 2021, PNAS). While I am typically hesitant to demand additional

experiments, the manuscript would be much stronger if the MD simulations were performed on a structure based on that of *Synechocystis*. H-bond networks in PSII are not identical between mesophilic and thermophilic cyanobacteria. Therefore, the presented MD data are not convincing.

We appreciate that the reviewer raised this important concern regarding the rationale behind our MD simulations. In our previous version of the manuscript, we used *T. vestitus* due to the recent availability of structural data on intermediates during the $S_3 \rightarrow [S_4] \rightarrow S_0$ transition (Bhowmick, et al. 2023, Nature). Nevertheless, *agreeing with this concern, we have approached extensive MD simulations for the PSII of the mesophilic Synechocystis PCC 6803, wildtype and N298A variant.* The initial structure of Photosystem II at 1.93 Å resolution was taken from the protein data bank (PDB ID: 7N8O, Gisriel, et al. 2021, PNAS). The major drawback of using the cryo-EM structure from *Synechocystis* PCC 6803 lies in the absence of structural data for S3. Therefore, to model the $S_3Y_z^{ox}$ state, the Mn_4CaO_6 cluster was taken from the crystallographically characterized PSII of *Thermosynechococcus vestitus* BP-1 (PDB ID: 8F4D), together with the C-terminus of D1, D1-Ala 344, which is “poorly resolved, and therefore modeled with low confidence” in the available high-resolution cryo-EM structure for *Synechocystis* PSII. Moreover, in these calculations the time period of the calculated trajectories was further extended, thereby improving the statistical significance of the calculation and the corresponding data analysis.

In the revised article, exclusively the results obtained for *Synechocystis* PCC 6803 are shown and discussed. The results do not differ strongly from the calculations previously done for the thermophilic PSII (see Supplementary Figure 20) but as pointed out by the reviewer, some differences in the amino acid sequence and H-bonding networks are present, which render the new calculation due to the exact species correspondence stronger.

A more minor point concerns the relationship between Figures 3 and 4. Please edit the text to connect these datasets more. As a non-FTIR expert, it is confusing why different wavenumbers are selected for discussion in the static data and for analysis in the time-resolved data.

A paragraph has been added where the selection of the wavenumbers for the time-resolved experiments is explained of the revised manuscript (2nd paragraph under headline “Time-resolved IR reveals biphasicity in the O₂-evolution transition of D1-N298A”, on pg. 10).

Three simple points:

Check the Chu, et al. citation in the caption of Table 1. Add that the WT O₂ rate in that work was 680 $\mu\text{mol}/\text{mg}/\text{h}$.

We have added this value to Table 1, alongside the rates determined in the here presented study. The numbers in the table are now marked to clearly indicate their origin (†Results from Chu, Nguyen and Debus³⁹. †Results (extrapolated) from Bao and Burnap⁴⁴. *Results from this study.)

Please tone down the usage of words such as astonishing, stunning, remarkable, etc. Using these terms more sparingly will increase their impact.

We agree. The wording has been tuned down; “astonishing” and “stunning” are no longer used; “remarkable” is used only once.

Figure S14 nicely shows distance fluctuations in the N298A mutant that are absent in WT. If the data are reproduced using the Synechocystis structure, consider moving this figure (or a smaller version of the figure) to the main text.

In the revised manuscript, we extended and improved the presentation of the MD simulation results. One panel of Figure 7 now displays an exemplary trajectory showing the distance fluctuations.

Reviewer #4 (Remarks to the Author):

Review report of manuscript “Three rate-determining protein roles in photosynthetic O₂-evolution addressed by time-resolved experiments on genetically modified photosystems”.

We thank the reviewer for comments which gain importance by highlighting the perception of an expert without long-standing experience in PSII research.

Mäusle and coworkers studied the time-evolved OEC complex by time resolved IR spectroscopy. The OEC evolving during water splitting problem has been attracted for many years and the long-standing question could be always the uniqueness of the refined model and parameters. The X-ray time resolved spectroscopy and diffraction would be another nice approach for the study, except the damage problem.

(a) We (the group of H. Dau) have studied PSII water oxidation repeatedly by time-resolved X-ray absorption spectroscopy since 2005 (cited ref. Haumann et al, 2005, Science). The thereby obtained results indeed represents an important basis for the interpretation of the time-resolved IR experiments of the present study. Another crucial basis of our study are the crystallographic models obtained by two research consortia as snapshots for various times after the exciting laser flash; the corresponding studies are cited repeatedly in our article. Addressing the here investigated PSII variants by time-resolved X-ray absorption or diffraction experiments is desirable as a future development, but may require several years for completion and clearly is beyond the scope of our present study.

The authors proposed to employ the TRIR spectroscopy for this study and it has been investigated by many groups in these years.

(b) TRIR spectroscopy on PSII is not as widespread assumed by the reviewer and far away from being routine experiments. These experiments still pose tremendous challenges in data collection and analysis. Inter alia, due to the weakness of the IR signals requiring a noise level (well) below 5 μ OD even for IR transients of ‘strong’ bands, extensive signal averaging is required; the need for long dark-adaptation periods before applying a laser flash sequence hinders repetitive experiments on the same PSII sample and also increases the amount of needed PSII samples drastically. Because of these challenges, to our best knowledge, aside from investigations by the author group (Dau and coworkers), there are only the time-resolved IR studies on PSII by Noguchi and coworkers, which are repeatedly cited.

However, the mutation of samples highlighted the novelty of this work. This manuscript contains important and interesting results, the novelty could make it publishable in Nature Comm. However, the current version needs to be refined/modified before the acceptance.

(c) We thank the reviewer for appreciating importance and relevance of our study. Our investigation indeed gains its strength by the combination of genetic protein variations (Debus and coworkers, Riverside), time-resolved O₂ polarography and IR spectroscopy (Dau and coworkers, Berlin), and computational chemistry (Guidoni and coworkers, L'Aquila). The article now has been extensively revised and further strengthened by additional analyses, including a significant extension of the MD calculations and clearer presentation of results in text and figures. We hope (and believe) that our revision addresses and clarifies the concerns expressed by all reviewers satisfactorily.

(1) The same question, the uniqueness of the mechanism proposed by authors. I would like to suggest the authors to do some test works, assuming different mechanism with different structural changes, to demonstrate the uniqueness of the model and parameters. The detailed descriptions can be listed in the revised SI and should add a short paragraph to describe in the discussion part of the manuscript.

As also suggested by Reviewer 1, the relation between experimental findings and their interpretation in terms of reaction/population kinetics (Figure 5a) and structural models (Figure 6) is presented more clearly in the revised manuscript. In particular, we have added a figure in the revised manuscript that shows simulations for the transient population of intermediate states (Figure 5) for the WT, D61A and N298A variant. We kindly ask the reviewer to read our related response to the suggestions of Reviewer 1.

The relation between the IR data and structural changes is not sufficiently well understood to relate IR spectra and transients directly to structural changes, unfortunately. Therefore, alternative models of specific structural changes cannot be tested based on the IR data. The IR data in the present study primarily serves the purpose to support the sequence of intermediates states shown in Figure 5a, with rate constants extracted from the time-resolved IR data.

(2). It could be useful if authors to plot the time-resolved IR spectra as a three-dimensional data, for instance, the x as the time, y axis as the lambda and the color bar, as the normal plotting of TA data. It would be useful to perform the global fitting to retrieve the lifetime components? More informative signals could obtain with decay associated spectrum and their decay timescales. The spectral profile could provide more valuable information, rather than the single-trace analysis.

The time-resolved data was collected at five selected wavenumbers, as explained in the revised manuscript in the first two paragraph under the headline "Time-resolved IR reveals biphasicity in the O₂-evolution transition of D1-N298A" (pg. 10).

The transients shown in Figure 4 indeed were simulated globally for each PSII variant (identical time constants in the transients collected at 1384, 1400, 1514 and 1544 cm⁻¹), as suggested by the reviewer. Since the data collection was restricted to four wavenumbers (plus 1478 cm⁻¹ shown in Supplementary Fig. 17), a 3D presentation was not approached.

(3). Do you think if this can be measured at lower temperature? The spectroscopic signals could be more informative, OK, maybe the flashing sample is hard to be achieved? Any idea to perform at low temperature? I knew the protein crystal and X-ray diffraction has been measured at lower temperature. For instance, they called it "chip" technology in Hamburg, which allows the researcher to diffract each crystal with a chip loading sample.

Going to low temperatures is an interesting suggestion. However, the here presented time-resolved experiments cannot be pursued at temperatures well below 0°C because crucial acceptor side as well as the donor side reactions will not only be slowed down but effectively blocked resulting in fast recombination of the initially created radical pairs or here irrelevant side reactions, e.g. formation chlorophyll or carotenoid triplets. (After decades of low-temperature EPR experiments, it is the special “beauty” of today’s time-resolved experiments temperature that they can track the functional PSII advancing through its reaction cycle directly at room temperature.)

(4). In Fig4, I never expect to have such amount of changes in signal, and even with the error bars. Or maybe I am not in the field for long time. Did the group perform in an advanced spectrometer or something wrong? Could you please explain it in details?

We apologize for the misleading y-axis scale in Fig. 4 (Fig. 3 in the revised manuscript). We are now using clear scale bars making it obvious that indeed very small absorption changes around 10 μ OD were detected.

(5). Except Fig. 1, Fig6 and Fig. 7, the rest of figures need to be refined. For instance, Fig. 3 looks quite mess. The authors need to consider to plot them in a better format, to meet the Nature Comm. Style.

We took the reviewer’s comment very seriously. Aside from Figure 1, all Figures have been revised for improved clarity and style. The especially harshly criticized Fig. 3 (“quite a mess”) has been moved to the Supplementary Material. We note that a similar presentation style as used in this figure has been used in several previous articles and previously has not been criticized as “quite a mess”, but we also see that it is a visually confusing figure in the context of the present article. The conclusions based on this complex figure are now stated in the main manuscript with reference to the Supplementary Figure 7, which we consider to be a good solution. This shift of the previous Fig. 3 to the SI also allowed us to complement the article by a further figure (Figure 5) with simulations of the time courses of intermediate populations.

(6). How about the X-ray diffraction? Is possible to perform these samples in the diffraction? Or electron diffraction.

Please see our response further above labeled by (a).

Point-by-point response to reviewer comments

There have been no further suggestions by Reviewer 2, 3 and 4.

Reviewer 1: "A couple of final comments regarding N298: is my understanding correct, that there is no expectation that the N298A mutation would affect the Y_z redox potential?"

This reviewer comment is addressed in the Discussion section by now referring explicitly to the Y_z redox potential (added statement in blue):

"The complete loss of this H-bond likely would affect the proton affinity (pK_a) of the H190-N_ε severely, **concomitantly the redox potential for Y_z oxidation**, and consequently the rate constant of the rate-determining proton-coupled ET in the S₃->S₄ transition. Experimentally this is not observed, which our MD simulations can explain .."

Reviewer 1: "And could the authors comment on the hypothesis of involvement of N298 in proton translocation (Chrysina et al. J. Phys. Chem. B 2019, 3068) or whether the EPR observations can be reinterpreted in light of the more likely participation of this residue in water insertion?"

This reviewer comment is addressed in the Introduction section by mentioning the option of a proton translocating role of N298, with citation of Chrysina et al. (ref. 33):

"Asparagine-298 (D1-N298) provides a link between Y_z (via its proton acceptor, D1-H190) and the water wheel (waters W26, W27, W28, W29, and W30) that has been proposed to serve as a water delivery site via the O1 channel^{17,19}; **tautomerization to its imidic acid form might support a proton translocating role of D1-N298**³³."

However, the EPR results of Chrysina et al. cannot be re-interpreted in light of our results, inter alia because the results of Chrysina et al. are not closely related to the oxygen evolution transition.